# Integrating Protein Dynamics into Structure-Based Drug Design via Full-Atom Stochastic Flows

**Xiangxin Zhou**[1,2*] **Yi Xiao**[3,4*] **Haowei Lin**[5] **Xinheng He**[6] **Jiaqi Guan**[7]
**Yang Wang**[4] **Qiang Liu**[2] **Feng Zhou**[4] **Liang Wang**[1,2] **Jianzhu Ma**[6,8†]
[1]School of Artificial Intelligence, University of Chinese Academy of Sciences
[2]New Laboratory of Pattern Recognition (NLPR),
 State Key Laboratory of Multimodal Artificial Intelligence Systems (MAIS),
 Institute of Automation, Chinese Academy of Sciences (CASIA)
[3]School of Information and Communication Technology, Griffith University
[4]Beijing StoneWise Technology Co Ltd
[5]Institute for Artificial Intelligence, Peking University
[6]Institute for AI Industry Research, Tsinghua University
[7]Department of Computer Science, University of Illinois Urbana-Champaign
[8]Department of Electronic Engineering, Tsinghua University

## Abstract

The dynamic nature of proteins, influenced by ligand interactions, is essential for comprehending protein function and progressing drug discovery. Traditional structure-based drug design (SBDD) approaches typically target binding sites with rigid structures, limiting their practical application in drug development. While molecular dynamics simulation can theoretically capture all the biologically relevant conformations, the transition rate is dictated by the intrinsic energy barrier between them, making the sampling process computationally expensive. To overcome the aforementioned challenges, we propose to use generative modeling for SBDD considering conformational changes of protein pockets. We curate a dataset of apo and multiple holo states of protein-ligand complexes, simulated by molecular dynamics, and propose a full-atom flow model (and a stochastic version), named DynamicFlow, that learns to transform apo pockets and noisy ligands into holo pockets and corresponding 3D ligand molecules. Our method uncovers promising ligand molecules and corresponding holo conformations of pockets. Additionally, the resultant holo-like states provide superior inputs for traditional SBDD approaches, playing a significant role in practical drug discovery.

## 1 Introduction

Modern deep learning is advancing several areas within drug discovery. Notably, among these, structure-based drug design (SBDD) (Anderson, 2003) emerges as a particularly significant and challenging domain. SBDD aims to discover drug-like ligand molecules specifically tailored to target binding sites. However, the complexity of chemical space and the dynamic nature of molecule conformations make traditional methods such as high throughput and virtual screenings inefficient. Additionally, relying on compound databases limits the diversity of identified molecules. Thus, deep generative models, such as autoregressive models (Luo et al., 2021; Peng et al., 2022) and diffusion models (Guan et al., 2023; Schneuing et al., 2022), have been introduced as a tool for *de novo* 3D ligand molecule design based on binding pockets, significantly transforming research paradigms.

However, most SBDD methods based on deep generative models assume that proteins are rigid (Peng et al., 2022; Guan et al., 2024). The dynamic behavior of proteins is crucial for practical drug discovery (Karelina et al., 2023; Boehr et al., 2009). Thermodynamic fluctuations result in proteins existing as an ensemble of various conformational states, and such states may interact with different drug molecules. The interconversion between these states often leads to the formation of cryptic

---

*Equal Contribution.
†Corresponding Author: Jianzhu Ma (majianzhu@tsinghua.edu.cn).

pockets, which are closely tied to the biological functions of the protein. During binding, the protein's structure may undergo fine-tuning, adopting different conformations to optimize its interaction with the drug, a phenomenon referred to as induced fit (Sherman et al., 2006). This binding can affect the protein's function by inhibiting the biological signaling pathway associated with a disease.

The underlying mechanism might involve blocking the binding of a natural substrate or shifting the protein's conformational distribution. For instance, kinases, which catalyze phosphorylation reactions by transferring phosphate groups from ATP to substrates, exist primarily in two conformations with different mainchain conformations: active (DFG-in) and inactive (DFG-out) (Panjarian et al., 2013). As illustrated in Fig. 1, the Abl kinase exists in both active and inactive apo states (6XR6, 6XR7), which differ in their DFG conformations. Type I kinase inhibitors, such as Dasatinib (2GQG), bind and stabilize the DFG-in conformation. In contrast, type II inhibitors like Imatinib (2HYY) stabilize the inactive conformation by occupying the ATP-binding pocket. Considering these dynamic conformational shifts is crucial for designing compounds that precisely target this kinase, as such structural nuances directly influence binding and efficacy.

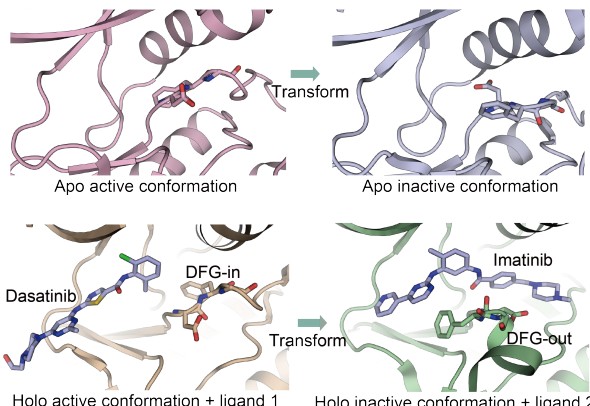

Figure 1: Comparison of Abl kinase domain conformations. In the top panel, the transition between the apo active and apo inactive conformations is shown. In the bottom panel, the active conformation with Dasatinib bound (DFG-in state) is compared to the inactive conformation with Imatinib bound (DFG-out state). The transformations between these states highlight the structural shifts critical for ligand binding.

We take a step towards integrating protein dynamics into structure-based drug design. First, we curate a dataset based on a large collection of molecular dynamics traces of protein-ligand complexes. Our dataset comprises both apo and multiple holo states of each protein-ligand complex. We propose an SE(3)-equivariant full-atom flow model, named DYNAMICFLOW, that simultaneously transforms the apo state of protein pockets and noisy ligands into holo states and their corresponding binding ligand molecules. Our method integrates insights from both machine learning and biology, utilizing the flow model's ability to transport one arbitrary distribution to another. We employ apo and holo state pairs as a natural coupling for conditional flow matching to train our flow model, making it learn the dynamics of both backbone frames and side-chain torsions of protein pockets. Simultaneously, we use continuous and discrete flow matching to model the distribution of atom positions, atom types, and bond types of ligand molecules. Additionally, we propose a stochastic version of our flow to further promote the robustness. Notably, to comprehensively capture the protein-ligand interaction, we meticulously design a full-atom model that incorporates both SE(3)-equivariant geometrical message passing layers and residue-level Transformer layers. The experimental results demonstrate the ability of our method to discover biologically meaningful protein conformation and promising ligand molecules. Interestingly, we find that the conformation states of protein pockets discovered by our method can serve as improved input for SBDD methods that treat proteins as rigid entities. We highlight our main contributions as follows:

- We present a meticulously curated dataset consisting of both apo and multiple holo states of protein-ligand complexes derived from molecular dynamics simulation, offering new opportunities for AI-driven drug discovery.

- We propose a full-atom flow model and a stochastic version that transform apo states of protein pockets to holo states and generates the corresponding 3D ligand molecules simultaneously.

- We design a full-atom model with both atom-level SE(3)-equivariant geometrical message passing layers and residue-level Transformer layers, which captures comprehensive protein-ligand interactions and flexibly models protein backbone frames and side-chain torsions.

- Our method generates meaningful protein conformation shifts and promising ligand molecules with high binding affinity. The discovered conformations also further enhance SBDD methods with rigid pockets as input.

## 2 RELATED WORK

**Structure-based Drug Design.** Structure-based drug design (SBDD) aims to develop ligand molecules capable of binding to specific protein targets. The advent of deep generative models has revolutionized the field, producing significant results. Luo et al. (2021); Peng et al. (2022); Liu et al. (2022a) and Zhang et al. (2022) employed an autoregressive model to generate atoms (and bonds) or fragments of a ligand molecule given a specific binding site with 3D structure. Guan et al. (2023); Schneuing et al. (2022); Lin et al. (2022) integrated diffusion models (Ho et al., 2020) into SBDD. These models initially generate the types and positions of atoms through iterative denoising, followed by determining bond types through post-processing. Some recent studies have sought to enhance these methods by incorporating biochemical prior knowledge. For an representative example, Huang et al. (2023) integrated protein-ligand interaction priors into both the forward and reverse processes to refine the diffusion models. While these studies have shown promising results in *in silico* evaluations, their disregard for protein dynamics limits their applicability in practical drug development. Please refer to App. L for extended related works.

**Generative Modeling of Protein Structures.** Deep generative models, especially diffusion models (Ho et al., 2020) (also known as score-based generative models (Song et al., 2021)) and flow models (Lipman et al., 2022; Liu et al., 2022b; Albergo & Vanden-Eijnden, 2022), have been widely utilized in modeling distributions of protein structures and attained impressive results. Protein structures are usually described with frames, which can be represented with elements of the Lie group SE(3), and side-chain torsion angles, which can be represented with elements of the Lie group SO(2) (Jumper et al., 2021; Watson et al., 2023). Both diffusion (De Bortoli et al., 2022; Huang et al., 2022) and flow (Chen & Lipman, 2023) models have been well extended to Riemannian manifolds, theoretically providing complete tools for modeling protein structures. These techniques have been extensively applied to various tasks related to protein structure modeling. Watson et al. (2023); Yim et al. (2023b;a); Bose et al. (2024) employed diffusion and flow models for protein backbone generation. Yim et al. (2024) and Lu et al. (2024a) further utilized these models for motif scaffolding and protein conformation sampling, respectively. Zhang et al. (2024a); Lee & Kim (2024) applied diffusion and flow models to protein side-chain packing. Plainer et al. (2023); Zhu et al. (2024); Huang et al. (2024b) proposed protein-ligand docking with flexibility on protein side-chain torsion under the framework of diffusion models. Lu et al. (2024b) considered flexibility on both protein backbone structure and side-chain torsion in protein-ligand docking. The above works inspire us to consider the complete degrees of freedom of proteins in the scenario of structure-based drug design.

## 3 METHOD

In this section, we will present our method, named DYNAMICFLOW, as illustrated in Fig. 2. In Section 3.1, we provide a clear statement about our problem setting and notations about SBDD considering protein dynamics along with a background on flow matching. In Section 3.2, we show the details about the training and inference of the full-atom flow, named DYNAMICFLOW-ODE, for this novel task. In Section 3.3, we show how to extend this flow model to a stochastic version, named DYNAMICFLOW-SDE, to improve its robustness. Finally, in Section 3.4, we introduced details about the multiscale full-atom architecture design of our flow model as shown in Fig. 3.

### 3.1 BACKGROUND AND PRELIMINARIES

We aim to generate a holo state of a protein pocket and its binding ligand molecule given its apo state. The ligand molecule can be represented as a set of $N_m$ atoms $\mathcal{M} = \{(\mathbf{x}_m^{(i)}, \mathbf{v}_m^{(i)}, \mathbf{b}_m^{(ij)})\}_{i,h \in \{1,...,N_m\}}$. The apo state of a protein pocket can be represented in atom level as a set of $N_p$ atoms $\mathcal{P}_0 = \{(\mathbf{x}_{p0}^{(i)}, \mathbf{v}_{p0}^{(i)})\}_{i=1}^{N_p}$. It can also be represented in residue level as a length-$D_p$ sequence of residues $\mathcal{S}_0 = [(\mathbf{a}_0^{(i)}, \mathbf{t}_0^{(i)}, \mathbf{r}_0^{(i)}, \chi_{00}^{(i)}, \chi_{10}^{(i)}, \chi_{20}^{(i)}, \chi_{30}^{(i)}, \chi_{40}^{(i)})]_{i=1}^{D_p}$. For the holo state, the first digits in all subscripts change from 0 to 1. Here $\mathbf{x} \in \mathbb{R}^3$ is the atom position, $\mathbf{v} \in \mathbb{R}^K$ is the atom type, $\mathbf{b} \in \mathbb{R}^B$ is the bond type, $\mathbf{a} \in \mathbb{R}^S$ is the amino acid type, $\mathbf{t} \in \mathbb{R}^3$ is the translation of the residue frame, $\mathbf{r} \in$ SO(3) is the rotation of the residue frame, $\chi_0 \in$ SO(2) is backbone torsion angle that governs the position of atom O, and $\chi_1, \chi_2, \chi_3, \chi_4 \in$ SO(2) are the side-chain torsion angles (see definitions in App. C). Note that $\mathbf{a}_0^{(i)} = \mathbf{a}_1^{(i)}$ and $\mathbf{v}_0^{(i)} = \mathbf{v}_1^{(i)}$ for all $i$ since protein conformation changes do not impact the amino acid types and atom types. The SBDD task, incorporating protein dynamics, can be formulated as modeling the conditional distribution $q(\mathcal{P}_1, \mathcal{M}|\mathcal{P}_0)$.

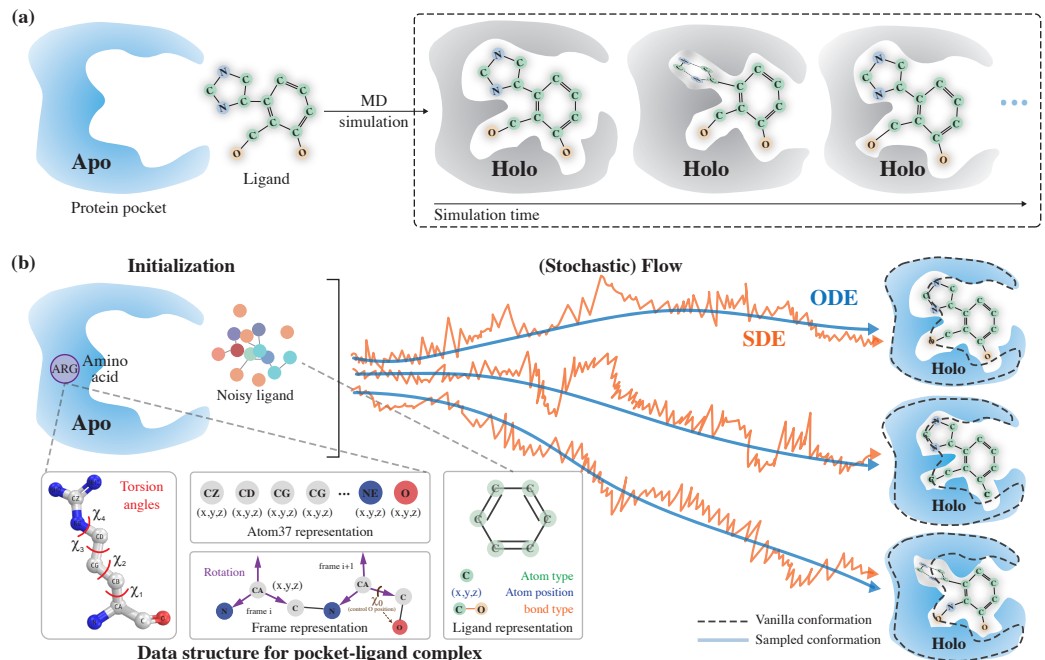

Figure 2: Overview of DYNAMICFLOW. (a) Our dataset consists of apo and multiple holo states of protein-ligand complexes derived from molecular dynamics simulation. (b) Our flow models, DYNAMICFLOW-ODE and DYNAMICFLOW-SDE, the generative process of ligand molecules along with the protein dynamics from apo to holo. The protein pocket is represented as both (i) residue frames and side-chain torsions and (ii) full atoms. The ligand molecule is represented as atom types, bond types, and atom positions.

We will use flow models to model the above conditional distribution. In the following, we provide a background on flow matching for modeling both continuous (Lipman et al., 2022; Liu et al., 2022b; Albergo & Vanden-Eijnden, 2022) and discrete (Campbell et al., 2024; Gat et al., 2024) variables, which we will use in the following sections.

We start with the continuous case. In flow matching, we consider a predefined *marginal probability path* $p_t$ that interpolates prior distribution $p_0$ and data distribution $p_1 = q$ to train a generative model that transports a source sample $\mathbf{x}_0 \sim p_0$ to a target sample $\mathbf{x}_1 \sim q_1$. We denote the joint distribution of $p_0$ and $p_1$ as $\pi(\mathbf{x}_0, \mathbf{x}_1)$ (also known as the data coupling). The probability path corresponds to a time-dependent flow $\psi_t : [0,1] \times \mathbb{R}^d \to \mathbb{R}^d$ and an associated vector field $u_t : [0,1] \times \mathbb{R}^d \to \mathbb{R}^d$, which can be defined via the ordinary differential equation (ODE) as $\dot{\mathbf{x}}_t = u_t(\mathbf{x}_t)$ where $\mathbf{x}_t = \psi_t(\mathbf{x}_0)$. One could use a neural network $v_\theta(\mathbf{x}_t, t)$ to regress the ground truth vector field $u_t(\mathbf{x}_t)$. However, the flow matching objective is intractable since we have no knowledge about the closed-form of the vector field $u_t$ that generates marginal distribution path $p_t$. The key insight (Lipman et al., 2022; Liu et al., 2022b) is to regress $v_\theta$ against the conditional vector field $u_t(\mathbf{x}_t|\mathbf{x}_0, \mathbf{x}_1)$, which generates the conditional probability path $p_t(\mathbf{x}_t|\mathbf{x}_0, \mathbf{x}_1)$, and use it to construct the target marginal path $p_t(\mathbf{x}_t) = \int_{\mathbf{x}_0, \mathbf{x}_1} p_t(\mathbf{x}_t|\mathbf{x}_0, \mathbf{x}_1)\pi(\mathbf{x}_0, \mathbf{x}_1)d\mathbf{x}_0 d\mathbf{x}_1$. The corresponding marginal vector field can also be constructed as $u_t(\mathbf{x}_t) := \int_{\mathbf{x}_0, \mathbf{x}_1} u_t(\mathbf{x}_t|\mathbf{x}_0, \mathbf{x}_1) \frac{p_t(\mathbf{x}_t|\mathbf{x}_0, \mathbf{x}_1)\pi(\mathbf{x}_0, \mathbf{x}_1)}{p_t(\mathbf{x}_t)}d\mathbf{x}_0 d\mathbf{x}_1$. The conditional flow matching loss is as follows:

$$\mathcal{L} = \mathbb{E}_{t \sim \mathcal{U}(0,1), \pi(\mathbf{x}_0, \mathbf{x}_1), p_t(\mathbf{x}_t|\mathbf{x}_0, \mathbf{x}_1)} \|v_\theta(\mathbf{x}_t, t) - u_t(\mathbf{x}_t|\mathbf{x}_1, \mathbf{x}_0)\|^2. \tag{1}$$

For modeling discrete variables, we follow the framework of Discrete Flow Matching (Campbell et al., 2024) which is based on Continuous-Time discrete Markov Chain (CTMC). Here $\mathbf{x} \in \{1, \ldots, S\}^D$ is a random discrete variable, e.g., a length-$D$ sequence where each element takes on one of $S$ states, and we denote $\boldsymbol{j}$ as a specific state of $\mathbf{x}$. We first introduce the concept of rate matrix $R_t \in \mathbb{R}^{S \times S}$, which plays a similar role to the vector field $u_t$ in the continuous case. The probability that $\mathbf{x}_t$ jumps to a different state $\boldsymbol{j}$ is $R_t(\mathbf{x}_t, \boldsymbol{j})dt$ after an infinitesimal time step $dt$ is $R(\mathbf{x}_t, \boldsymbol{j})dt$, i.e., $p_{t+dt|t}(\boldsymbol{j}|\mathbf{x}) = \delta\{\mathbf{x}, \boldsymbol{j}\} + R_t(\mathbf{x}_t, \boldsymbol{j})dt$ where $\delta\{\boldsymbol{i}, \boldsymbol{j}\}$ is an element-wise Kronecker delta which 1 in dimension $d$ when $\boldsymbol{i}^d = \boldsymbol{j}^d$ and 0 otherwise. To

build a discrete flow, we also use the conditional probability path to construct the marginal one as $p_t(\mathbf{x}_t) := \mathbb{E}_{p(x_1)}[p_t(\mathbf{x}_t|\mathbf{x}_1)]$. We define $R_t(\mathbf{x}_t, \boldsymbol{j}|\mathbf{x}_1)$ as a conditional rate matrix that generates $p_t(\mathbf{x}_t|\mathbf{x}_1)$. Notably, $R_t(\mathbf{x}_t, \boldsymbol{j}|\mathbf{x}_1)$ can usually be in a simple formula. For example, it can be that $R_t(\mathbf{x}_t, \boldsymbol{j}|\mathbf{x}_1) := \delta\{\mathbf{x}_1, \boldsymbol{j}\}\delta\{\mathbf{x}_t, M\}/(1-t)$ where $M$ is the mask token. It can be proved that the marginal rate matrix $R_t(\mathbf{x}_t, \boldsymbol{j}) := \mathbb{E}_{p(\mathbf{x}_1|\mathbf{x}_t)}[R_t(\mathbf{x}_t, \boldsymbol{j}|\mathbf{x}_1)]$ corresponds to the marginal probability path $p_t(\mathbf{x}_t)$ which we have defined above, where $p(\mathbf{x}_1|\mathbf{x}_t) = p_t(\mathbf{x}_t|\mathbf{x}_1)q(\mathbf{x}_1)/p_t(\mathbf{x}_t)$. We can use a neural network $p_t^{\theta}(\mathbf{x}_1|\mathbf{x}_t)$ to approximate the posterior $p_t(\mathbf{x}_1|\mathbf{x}_t)$. We denote $R_t^{\theta}(\mathbf{x}_t, \cdot) := \mathbb{E}_{p_t^{\theta}(\mathbf{x}_1|\mathbf{x}_t)}[R_t(\mathbf{x}_t, \cdot|\mathbf{x}_1)]$. We can generate a sample by iteratively sampling from the process as

$$p_{t+dt}(\mathbf{x}_{t+dt}|\mathbf{x}_t) = \delta\{\mathbf{x}_{t+dt}, \mathbf{x}_t\} + R_t^{\theta}(\mathbf{x}_t, \mathbf{x}_{t+dt})dt + o(dt). \tag{2}$$

### 3.2 FULL-ATOM FLOW FOR SBDD WITH PROTEIN DYNAMICS

One advantage of flow models is their ability to transport any arbitrary distribution to another, without imposing constraints on the source or target distribution. Fully leveraging this advantage to model the conditional distribution $q(\mathcal{P}_1, \mathcal{M}|\mathcal{P}_0)$, we build a flow, named DYNAMICFLOW-ODE, that transforms apo to holo states and generates ligand molecules from a noisy prior distribution simultaneously. The apo state $\mathcal{P}_0$ and holo state $\mathcal{P}_1$ naturally serve as samples from the source and target distribution for flow matching. Formally, we define the source distribution $p_0$ as

$$p_0(\mathcal{P}, \mathcal{M}|\mathcal{P}_0) := \delta(\mathcal{P}, \mathcal{P}_0) \prod_{i=1}^{N_m} \mathcal{N}(\mathbf{x}_m^{(i)}; \mathrm{CoM}(\mathcal{P}_0), \boldsymbol{I})\delta\{\mathbf{v}_m^{(i)}, M_v\} \prod_{i,j=1}^{N_m} \delta\{\mathbf{b}_m^{(ij)}, M_b\}, \tag{3}$$

where $\delta(\cdot, \cdot)$ is the Dirac delta function, $\delta\{\cdot, \cdot\}$ is the discrete Kronecker delta function, $\mathrm{CoM}(\cdot)$ is the function that calculates center of mass, and $M_v$ (resp. $M_b$) is the token for atom (resp. bond) type. This means that the pocket directly starts with the apo state, atom/bond types of ligand molecules start with mask tokens, and atom positions of molecules start with a normal distribution around the CoM of the apo pocket. The target distribution $p_1(\mathcal{P}, \mathcal{M}|\mathcal{P}_0)$ is the data distribution $q$. Specifically, we use the multiple holo states and binding ligand molecules that correspond to apo state $\mathcal{P}_0$ in our training dataset as the data coupling $\pi(\mathcal{M}_0, \mathcal{P}_0, \mathcal{M}_1, \mathcal{P}_1|\mathcal{P}_0)$ and use the predefined conditional probability path $p_t(\mathcal{M}_t, \mathcal{P}_t|\mathcal{M}_0, \mathcal{P}_0, \mathcal{M}_1, \mathcal{P}_1)$ to sample "interpolant" to train our flow model.

Next, we show how to derive the vector field $u_t$ for the continuous variables (i.e., pocket residues' translation, rotation, torsion angles, and ligand molecules' atom positions). Both pocket residues' translation $\mathbf{t}$ and ligand molecules' atom positions $\mathbf{x}_m$ live in 3D Euclidean space. Taking $\mathbf{t}$ as an example, given samples from source and target distribution, i.e., $\mathbf{t}_0$ and $\mathbf{t}_1$, we define the conditional probability path as the linear interpolant between these two samples and derive the conditional vector field as follows ($t$ in superscript stands for "translation" and $t$ in subscript stands for "time"):

$$p_t^t(\mathbf{t}_t^{(i)}|\mathbf{t}_0^{(i)}, \mathbf{t}_1^{(i)}) := \delta(\mathbf{t}_t^{(i)}, t\mathbf{t}_1^{(i)} + (1-t)\mathbf{t}_0^{(i)}) \quad \text{and} \quad u_t^t(\mathbf{t}_t^{(i)}|\mathbf{t}_1^{(i)}, \mathbf{t}_0^{(i)}) = \mathbf{t}_1^{(i)} - \mathbf{t}_0^{(i)} = \frac{\mathbf{t}_1^{(i)} - \mathbf{t}_0^{(i)}}{1-t}. \tag{4}$$

Instead of directly predicting the vector field, we use $\mathbf{t}_{\theta}(\mathbf{t}_t^{(i)}, t)$ to predict the "clean" sample at time 1 and express the predicted vector field as $v_{\theta}(\mathbf{t}_t^{(i)}, t) := (\mathbf{t}_{\theta}(\mathbf{t}_t^{(i)}, t) - \mathbf{t}_0^{(i)})/(1-t)$. Please note that while the complete context $(\mathcal{M}_t, \mathcal{P}_t)$ is indeed input into the network parameterized with $\theta$, we omit it here for simplicity. This convention is maintained throughout the entire paper. The conditional flow matching (CFM) objective for translation of residue frame at time $t$ can be formulated as:

$$\mathcal{L}_t^t = \mathbb{E}_{\pi(\mathbf{t}_0, \mathbf{t}_1), p_t^t(\mathbf{t}_t^{(i)}|\mathbf{t}_0^{(i)}, \mathbf{t}_1^{(i)})} \big\| v_{\theta}(\mathbf{t}_t^{(i)}, t) - (\mathbf{t}_1^{(i)} - \mathbf{t}_0^{(i)}) \big\|_2^2. \tag{5}$$

For the rotation of residue frames, we use the geodesic interpolant to define the conditional probability path and conditional vector path as follows:

$$p_t^r(\mathbf{r}_t^{(i)}|\mathbf{r}_0^{(i)}, \mathbf{r}_1^{(i)}) := \delta\big(\mathbf{r}_t^{(i)}, \exp_{\mathbf{r}_0^{(i)}}\big(t\log_{\mathbf{r}_0^{(i)}}\big(\mathbf{r}_1^{(i)}\big)\big)\big) \quad \text{and} \quad u_t^r(\mathbf{r}_t^{(i)}|\mathbf{r}_0^{(i)}, \mathbf{r}_1^{(i)}) = \frac{\log_{\mathbf{r}_t^{(i)}}(\mathbf{r}_1^{(i)})}{1-t}. \tag{6}$$

Similar to the case for translation, we use $\mathbf{r}_{\theta}(\mathbf{r}_t^{(i)}, t)$ to predict the "clean" sample at time 1 and express the predicted vector field on SO(3) manifold as $v_{\theta}(\mathbf{r}_t^{(i)}, t) := \log_{\mathbf{r}_t^{(i)}}(\mathbf{r}_{\theta}(\mathbf{r}_t^{(i)}, t))/(1-t)$. The CFM objective for rotation of residue frame at time $t$ can be formulated as:

$$\mathcal{L}_t^r = \mathbb{E}_{\pi(\mathbf{r}_0, \mathbf{r}_1), p_t^t(\mathbf{r}_t^{(i)}|\mathbf{r}_0^{(i)}, \mathbf{r}_1^{(i)})} \bigg\| v_{\theta}(\mathbf{r}_t^{(i)}, t) - \frac{\log_{\mathbf{r}_t^{(i)}}(\mathbf{r}_1^{(i)})}{1-t} \bigg\|_{\mathrm{SO}(3)}^2. \tag{7}$$

Torsion angles of protein pockets lie on high-dimensional tori. Note the slight abuse of notation that we use $\mathcal{M}$ to represent both the manifold and the molecule. For a torus $\mathcal{M} := [0, k\pi]$ $(k = 1, 2)$, the exponential and logarithm maps are as follows Chen & Lipman (2023):

$$\exp_u(x) = (x + u)\%(k\pi) \quad \text{and} \quad \log_x(y) = \arctan2(\sin(y - x), \cos(y - x)), \tag{8}$$

where $x, y \in \mathcal{M}$ and $u \in \mathcal{T}_x\mathcal{M}$. Following Li et al. (2024), we define $\text{wrap}(u) := (u+\pi)\%(2\pi) - \pi$. While we model 5 torsion angles, here we use $\chi$ for brevity. The conditional probability path and conditional vector field can be defined as follows:

$$p_t^\chi(\chi_t^{(i)}|\chi_0^{(i)}, \chi_t^{(i)}) := \delta\left(\chi_t^{(i)}, \left(\chi_0^{(i)} + t\text{wrap}(\chi_1^{(i)} - \chi_0^{(i)})\right)\%(k\pi)\right), \tag{9}$$

$$u_t(\chi_t^{(i)}|\chi_0^{(i)}, \chi_1^{(i)}) = \text{wrap}(\chi_1^{(i)} - \chi_0^{(i)}) = \frac{\text{wrap}(\chi_1^{(i)} - \chi_t^{(i)})}{1 - t}. \tag{10}$$

Here we introduce $k = 1$ to handle the special cases where some side-chain torsion angles of certain residues exhibit $\pi$-rotation-symmetry (e.g., $\chi_2$ of ASP). For other general cases, $k = 2$. Refer to App. C for details. We use $\chi_\theta(\chi_t^{(i)}, t)$ to predict the "clean" sample at time 1 and express the predicted vector field on tori as $v_\theta(\chi_t^{(i)}, t) := \text{wrap}(\chi_\theta(\chi_t^{(i)}, t) - \chi_0^{(i)})/(1 - t)$. Also following Li et al. (2024), we use the flat metric on tori in CFM objective as follows:

$$\mathcal{L}_t^\chi = \mathbb{E}_{\pi(\chi_0, \chi_1), p_t^t(\chi_t^{(i)}|\chi_0^{(i)}, \chi_1^{(i)})}\left\|\text{wrap}\left(v_\theta(\chi_t^{(i)}, t) - \frac{\chi_1^{(i)} - \chi_0^{(i)}}{1 - t}\right)\right\|^2. \tag{11}$$

So far, We have now covered all cases of the continuous variables. The sampling (i.e., inference) procedure of these continuous variables is solving ODEs defined on the corresponding space.

Next, we show how we model discrete variables, i.e., atom types and bond types of ligand molecules. We take bond types $\mathbf{b}_m^{(ij)}$ of ligand molecules as an example. In the following, we omit the subscript $m$ that stands for "molecule" for brevity without ambiguity. We define the conditional probability path and the conditional rate matrix as follows:

$$p_t^b(\mathbf{b}_t^{(ij)}|\mathbf{b}_0^{(ij)}, \mathbf{b}_1^{(ij)}) := \text{Cat}(t\delta\{\mathbf{b}_1^{(ij)}, \mathbf{b}_t^{(ij)}\} + (1 - t)\delta\{\mathbf{b}_t^{(ij)}, M\}), \tag{12}$$

$$R_t^b(\mathbf{b}_t^{(ij)}, n|\mathbf{b}_1^{(ij)}) = \frac{\delta\{\mathbf{b}_1^{(ij)}, n\}\delta\{\mathbf{b}_t^{(ij)}, M\}}{1 - t}, \tag{13}$$

where $n \in \{1, \ldots, B\}$ is a specific bond type. A straightforward interpretation of the above conditional discrete flow is that we linearly interpolate between a probability mass concentrated entirely on the mask token and the data distribution. As introduced in Section 3.1, we train a neural network $p_\theta^b(\mathbf{b}_1^{(ij)}|\mathbf{b}_t^{(ij)})$ with simple cross-entropy loss as follows:

$$\mathcal{L}_t^b = \mathbb{E}_{\pi(\mathbf{b}_0, \mathbf{b}_1), p_t^b(\mathbf{b}_t^{(ij)}|\mathbf{b}_0^{(ij)}, \mathbf{b}_1^{(ij)})}[-\log p_\theta^b(\mathbf{b}_1^{(ij)}|\mathbf{b}_t^{(ij)})]. \tag{14}$$

The estimated marginal rate matrix can be expressed as

$$R_\theta^b(\mathbf{b}_t^{(ij)}, n) = \mathbb{E}_{p_\theta^b(\mathbf{b}_1^{(ij)}|\mathbf{b}_t^{(ij)})}[R_t^b(\mathbf{b}_t^{(ij)}, n|\mathbf{b}_1^{(ij)})] = \frac{p_\theta^b(\mathbf{b}_1^{(ij)}|\mathbf{b}_t^{(ij)})}{1 - t}\delta\{\mathbf{b}_t^{(ij)}, M\}. \tag{15}$$

During sampling, the transition step from $t$ to $t + dt$ is

$$p_{t+dt}(n|\mathbf{b}_t^{(ij)}) = \delta\{\mathbf{b}_t^{(ij)}, n\} + R_\theta^b(\mathbf{b}_t^{(ij)}, n)dt. \tag{16}$$

To better capture interactions between atoms, we propose an interaction loss that directly regresses the distances between predicted atom pairs and the ground-truth distances in local space. For time $t$, we denote the predicted "clean" holo state and ligand molecule as

$$\hat{\mathcal{S}}_1 := [(\mathbf{a}_0^{(i)}, \mathbf{t}_\theta^{(i)}, \mathbf{r}_\theta^{(i)}, \chi_{1\theta}^{(i)}, \chi_{2\theta}^{(i)}, \chi_{3\theta}^{(i)}, \chi_{4\theta}^{(i)})]_{i=1}^{D_p} \quad \text{and} \quad \hat{\mathcal{M}}_1 := \{(\hat{\mathbf{x}}_{m1}^{(i)}, \hat{\mathbf{v}}_{m1}^{(i)})\}_{i=1}^{N_p}, \tag{17}$$

where we denote $\mathbf{r}_\theta(\mathbf{r}_t^{(i)}, t)$ as $\mathbf{t}_\theta^{(i)}$ for brevity and similarly for other variables. Note that here we omit the bond types in $\hat{\mathcal{M}}_1$ because we do not use it in the interaction loss. The frame representation is then converted to atom representation $\hat{\mathcal{P}}_1 := \{(\hat{\mathbf{x}}_{p1}^{(i)}, \hat{\mathbf{v}}_{p1}^{(i)})\}_{i=1}^{N_p} = \text{FrameToAtom}(\hat{\mathcal{S}}_1)$. See details about FrameToAtom$(\cdot)$ in App. B. We denote all predicted atoms in the protein-ligand complex as $\hat{\mathcal{A}}_1 := \{(\hat{\mathbf{x}}_1^{(i)}, \hat{\mathbf{v}}_1^{(i)})\}_{i=1}^{N_p+N_m} = \hat{\mathcal{P}}_1 \cup \hat{\mathcal{M}}_1$ and the corresponding ground truth as $\mathcal{A}_1 := \{(\mathbf{x}_1^{(i)}, \mathbf{v}_1^{(i)})\}_{i=1}^{N_p+N_m} = \mathcal{P}_1 \cup \mathcal{M}_1$. The interaction loss can be formulated as follows:

$$\mathcal{L}_t^{\text{int}} = \sum_{i=1}^{N_p+N_m}\sum_{j=i+1}^{N_p+N_m}\|\hat{d}_1^{(ij)} - d_1^{(ij)}\|^2 \cdot \mathbb{1}\{d_1^{(ij)} \leq 3.5\} \cdot \mathbb{1}\{t > 0.65\}, \tag{18}$$

where $\hat{d}_1^{(ij)} = \|\hat{\mathbf{x}}_1^{(i)} - \hat{\mathbf{x}}_1^{(j)}\|$, $d_1^{(ij)} = \|\mathbf{x}_1^{(i)} - \mathbf{x}_1^{(j)}\|$, and $\mathbb{1}\{\cdot\}$ is the indicator function.

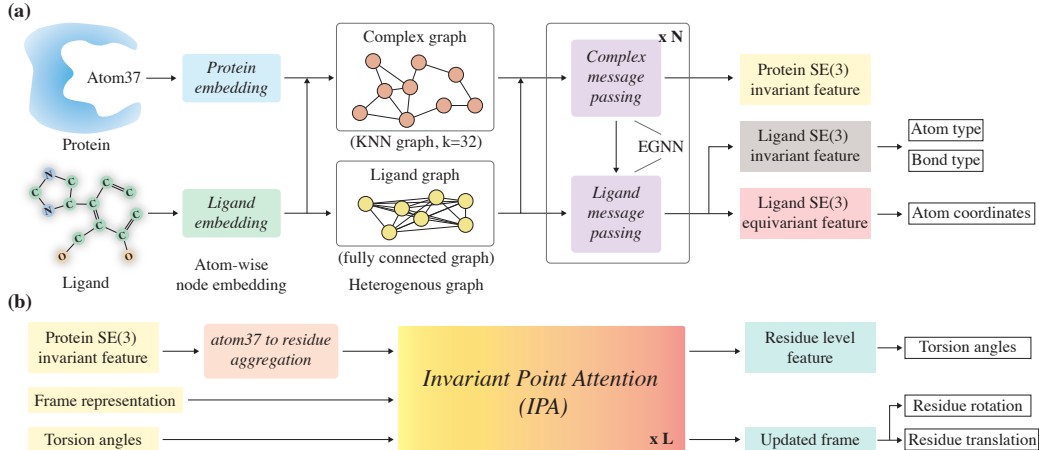

Figure 3: Illustration of our multiscale full-atom model architecture: (a) atom-level SE(3)-equivariant graph neural network; (b) residue-level Transformers.

## 3.3 EXTENSION TO STOCHASTIC FULL-ATOM FLOW

The probability path induced by linear interpolant between apo and holo state of pocket has limited set-theoretic support, thus potentially leading to inferior robustness. This motivates us to build a stochastic flow, named DYNAMICFLOW-SDE, since SDEs offer the significant advantage of being more robust to noise in high-dimensional spaces compared to ODEs (Tong et al., 2023; Shi et al., 2023; Liu et al., 2023). Based on the way that we build DYNAMICFLOW-ODE as we introduced in Section 3.2, we can easily extend to SDEs via simply updating the conditional probability paths in training and sampling. Bose et al. (2024) offers theoretical justification that conditional flow matching loss and original flow matching loss remain equivalent with respect to optimization.

For variables lie in Euclidean space, i.e., the translation of residue frames $\mathbf{t}^{(i)}$ and the atom positions $\mathbf{a}^{(i)}$, we can update the conditional probability path from Eq. (4) to:

$$p_t^t(\mathbf{t}_t^{(i)}|\mathbf{t}_0^{(i)}, \mathbf{t}_1^{(i)}) := \mathcal{N}(\mathbf{t}_t^{(i)}; t\mathbf{t}_1^{(i)} + (1-t)\mathbf{t}_0^{(i)}, \gamma^2 t(1-t)\boldsymbol{I}), \tag{19}$$

where $\gamma$ is a constant hyperparameter that controls the noise scale.

For variables lie on SO(3) manifold, i.e., the rotation of residue frames $\mathbf{r}^{(i)}$, we update the conditional probability path in Eq. (6) to an efficient simulation-free approximation of correct conditional stochastic flow on SO(3) suggested by Bose et al. (2024):

$$p_t^r(\mathbf{r}_t^{(i)}|\mathbf{r}_0^{(i)}, \mathbf{r}_t^{(i)}) := \mathcal{IG}_{\text{SO(3)}}(\mathbf{r}_t^{(i)}; \exp_{\mathbf{r}_0^{(i)}}(t\log_{\mathbf{r}_0^{(i)}}(\mathbf{r}_1^{(i)})), \gamma t(1-t)), \tag{20}$$

where $\mathcal{IG}_{\text{SO(3)}}$ denotes the isotropic Gaussian distribution on SO(3) (Nikolayev & Savyolov, 1900).

For variables lie on tori, i.e., the torsion angles of residues $\chi^{(i)}$, we update the conditional probability path in Eq. (10) to:

$$p_t^\chi(\chi_t^{(i)}|\chi_0^{(i)}, \chi_t^{(i)}) \propto \sum_{d \in \mathbb{Z}} \exp\left(-\frac{\left\|\chi_t^{(i)} - (\chi_0^{(i)} + t\text{wrap}(\chi_1^{(i)} - \chi_0^{(i)}))\%(k\pi) + k\pi d\right\|^2}{2\gamma^2 t(t-1)}\right), \tag{21}$$

which is a wrapped normal distribution. This can be sampled directly (Jing et al., 2022).

## 3.4 MULTISCALE FULL-ATOM MODEL ARCHITECTURE

Our flow models employ a multiscale scheme as shown in Fig. 3, which is comprised of two main parts: (a) atom-level SE(3)-equivariant graph neural network; (b) residue-level Transformers.

In the atom-level GNN, inspired by Guan et al. (2024), we maintain both node-level and edge-level hidden states in the neural network by modifying widely used EGNN (Satorras et al., 2021). During training, for time $t$, we build a $k$-nearest neighbors (knn) graph $G_c$ upon the atoms of the protein-ligand complex to capture the protein-ligand interaction:

$$\Delta\mathbf{h}_c^{(i)} \leftarrow \sum_{j \in \mathcal{N}_c(i)} \phi_c(\mathbf{h}^{(i)}, \mathbf{h}^{(j)}, \|\mathbf{x}^{(i)} - \mathbf{x}^{(j)}\|, E^{(ij)}, t), \tag{22}$$

where $\mathbf{h}$ is the atom-level hidden state, $\mathcal{N}_c(i)$ is the neighbors of $i$ in $\mathcal{G}_c$, $E^{(ij)}$ indicates the edge $ij$ is a protein-protein, ligand-ligand, or protein-ligand edge. We also build a fully connected ligand graph $G_m$ upon all ligand atoms to model the chemical interaction inside the ligand molecule:

$$\mathbf{m}^{(ij)} \leftarrow \phi_m(\|\mathbf{x}^{(i)} - \mathbf{x}^{(j)}\|, \mathbf{e}^{(ij)}) \quad \text{and} \quad \Delta\mathbf{h}_m^{(i)} \leftarrow \sum_{j \in \mathcal{N}_m(i)} \phi_m(\mathbf{h}^{(i)}, \mathbf{h}^{(j)}, \mathbf{m}^{(ji)}, t), \quad (23)$$

where $\mathbf{e}$ is the bond-level hidden state. We then update the atom and bond's hidden state as follows:

$$\mathbf{h}^{(i)} \leftarrow \mathbf{h}^{(i)} + \phi_h(\Delta\mathbf{h}_c^{(i)} + \Delta_m^{(i)}), \quad (24)$$

$$\mathbf{e}^{(ji)} \leftarrow \sum_{k \in \mathcal{N}_m(j) \setminus \{i\}} \phi_e(\mathbf{h}^{(i)}, \mathbf{h}^{(j)}, \mathbf{h}^{(k)}, \mathbf{m}^{(jk)}, \mathbf{m}^{(ji)}, t), \quad (25)$$

Then we update the SE(3)-equivariant features (i.e. positions of ligand molecules) as:

$$\Delta\mathbf{x}_c^{(i)} \leftarrow \sum_{j \in \mathcal{N}_c(i)} \mathbf{d}^{(ji)} \phi_c^x(\mathbf{h}^{(i)}, \mathbf{h}^{(j)}, d^{(ij)}, t), \quad \Delta\mathbf{x}_m^{(i)} \leftarrow \sum_{j \in \mathcal{N}_m(i)} \mathbf{d}^{(ji)} \phi_m^x(\mathbf{h}^{(i)}, \mathbf{h}^{(j)}, d^{(ji)}, \mathbf{m}^{(ji)}, t) \quad (26)$$

where $\mathbf{d}^{(ji)} = \mathbf{x}^{(j)} - \mathbf{x}^{(i)}$ and $d^{(ij)} = \|\mathbf{d}^{(ji)}\|$. We only update ligand atom positions in the atom-level part as $\mathbf{x}^{(i)} \leftarrow \mathbf{x}^{(i)} + (\Delta\mathbf{x}_c^{(i)} + \Delta\mathbf{x}_m^{(i)}) \cdot \mathbb{1}_m(i)$, where $\mathbb{1}_m(i)$ is an indicator of ligand atom, and leave the pocket update in the residue-level part. The output equivariant (resp. invariant) hidden states of ligand nodes are used to predict atom positions (resp. atom/bond types) of ligand molecules.

We aggregate the final protein atom hidden states $\mathbf{h}$ output by the above full-atom GNN into residue level according to the atom37 template (Jumper et al., 2021). The aggregated residue-level hidden states along with amino acid type $\mathbf{a}$, 5 torsion angles $\chi$, and frame $(\mathbf{t}, \mathbf{r})$ are input into a residue-level Transformer model which is composed of node embedding layer, edge embedding layer, multiple Invariant Point Attention (IPA) blocks, following Jumper et al. (2021); Yim et al. (2023b). In each IPA block, residue-level hidden state $\mathbf{h}$ and frame $(\mathbf{t}, \mathbf{r})$ are updated. The final updated frames are used as predictions. Torsion angles are predicted based on the final residue-level hidden states.

## 4 EXPERIMENTS

**Data Curation.** We curate our dataset based on MISATO dataset (Siebenmorgen et al., 2024). The original dataset contains approximately 20,000 protein-ligand complexes with associated 8ns molecular dynamics (MD) simulation trajectories. We filter out complexes where the ligands are oligopeptides, yielding 12,695 complexes for further processing. For each complex, we efficiently use the data by clustering the holo-ligand conformations with an RMSD threshold of 1.0 Å. Finally, to filter out the invalid MD simulation results, we select complexes whose averaged RMSD between the ligand conformations simulated by MD and native ligand structures is less than 3 Å. The above procedure results in a dataset containing 5,692 complexes with 46,235 holo-ligand conformations and corresponding apo structures. More details can be found in App. A.

**Baselines.** We compare our models with three representative baselines for SBDD: **Pocket2Mol** (Peng et al., 2022) generate 3D molecules by *sequentially* placing atoms around a given protein pocket; **TargetDiff** (Guan et al., 2023) generates atom coordinates and atom types based on a diffusion model and bonds are determined with a post-processing algorithm; **IPDiff** (Huang et al., 2023) incorporated protein-ligand interaction priors into both the forward and reverse processes to enhance the diffusion models. Since diffusion models require the number of atoms in the ligand molecule as input, we align TargetDiff and IPDiff with our methods based on the number of ligand atoms, denoting them as **TargetDiff\*** and **IPDiff\***. We also compare our methods on the ability of pocket conformation sampling with two baselines: a simple method that injected appropriate noise into

Table 1: Summary of properties of reference molecules and molecules generated by our model and other baselines. ($\uparrow$) / ($\downarrow$) denotes a larger / smaller number is better.

| | Vina Score ($\downarrow$) | QED ($\uparrow$) | SA ($\uparrow$) | Linpiski ($\uparrow$) | logP | High Affi. ($\uparrow$) | Comp. Rate ($\uparrow$) |
|---|---|---|---|---|---|---|---|
| Reference | -7.84±3.11 | 0.53±0.19 | 0.67±0.15 | 4.45±0.77 | 1.64±2.17 | N/A | N/A |
| Pocket2Mol | -5.50±1.25 | 0.54±0.12 | 0.83±0.10 | 1.70±1.52 | 4.98±0.14 | 28.6% | N/A |
| TargetDiff | -5.09±4.81 | 0.42±0.18 | 0.53±0.13 | 3.20±2.39 | 4.32±0.81 | 44.0% | 68.2% |
| TargetDiff* | -5.53±4.09 | 0.39±0.21 | 0.55±0.15 | 3.02±2.75 | 4.11±1.10 | 45.7% | 62.4% |
| IPDiff | -7.55±5.59 | 0.38±0.17 | 0.50±0.14 | 4.10±0.77 | 5.07±2.94 | 51.0% | 67.8% |
| IPDiff* | -6.23±5.26 | 0.40±0.20 | 0.55±0.15 | 3.94±1.00 | 5.04±3.34 | 49.5% | 67.3% |
| DYNAMICFLOW-ODE | -7.28±1.98 | 0.53±0.20 | 0.61±0.14 | 3.13±2.62 | 4.45±0.81 | 51.0% | 70.2% |
| DYNAMICFLOW-SDE | -7.65±1.59 | 0.53±0.15 | 0.53±0.17 | 4.34±2.58 | 4.25±0.90 | 52.5% | 73.6% |

Table 2: Performance of SBDD methods with rigid pocket inputs on apo/our pockets, where "our pockets" refers to the pocket structures generated by our method.

| | Vina Score (↓) | QED (↑) | SA (↑) | Linpiski (↑) | logP | High Affi. (↑) | Comp. Rate (↑) |
|---|---|---|---|---|---|---|---|
| Pocket2Mol | -5.50±1.25 | 0.54±0.12 | 0.83±0.10 | 1.70±1.52 | 4.98±0.14 | 28.6% | N/A |
| Pocket2Mol + Our Pocket | -5.92±0.88 | 0.57±0.07 | 0.88±0.10 | 2.09±1.24 | 5.00±0.00 | 28.6% | N/A |
| TargetDiff | -5.09±8.81 | 0.42±0.18 | 0.53±0.13 | 3.20±2.39 | 4.32±0.81 | 44.0% | 68.2% |
| TargetDiff + Our Pocket | -9.00±3.19 | 0.44±0.18 | 0.53±0.14 | 3.82±2.29 | 4.22±0.91 | 61.2% | 79.6% |
| TargetDiff* | -5.53±4.09 | 0.39±0.21 | 0.55±0.15 | 3.02±2.75 | 4.11±1.10 | 45.7% | 62.4% |
| TargetDiff* + Our Pocket | -7.82±3.50 | 0.48±0.22 | 0.58±0.14 | 3.65±2.87 | 4.14±1.09 | 51.0% | 79.72% |
| IPDiff | -7.55±5.59 | 0.38±0.17 | 0.50±0.14 | 4.10±0.77 | 5.07±2.94 | 51.0% | 67.8% |
| IPDiff + Our Pocket | -11.04±3.76 | 0.40±0.16 | 0.46±0.15 | 3.84±0.74 | 6.97±3.33 | 83.7% | 83.4% |
| IPDiff* | -6.23±5.26 | 0.40±0.20 | 0.55±0.15 | 3.94±1.00 | 5.04±3.34 | 49.5% | 67.3% |
| IPDiff* + Our Pocket | -9.71±3.66 | 0.42±0.16 | 0.52±0.13 | 4.00±0.83 | 5.92±3.67 | 81.6% | 82.1% |

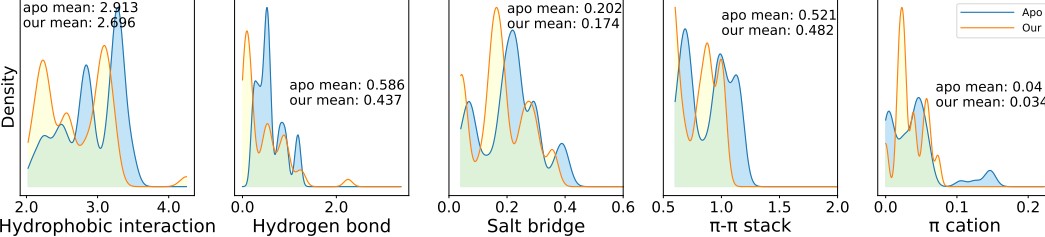

Figure 4: Distribution of differences in protein-ligand non-covalent interaction numbers of apo/our pockets and ligands designed by TargetDiff from those of ground-truth holo-ligand complexes.

apo structures; **Str2Str** (Lu et al., 2024a) that repurposed the protein backbone generation model, FrameDiff (Yim et al., 2023b), for protein conformation sampling by first injecting noise into apo structures and then denoising by diffusion models.

**Evaluation.** We select 50 complexes that have no overlap with the training set as the test set. We evaluate both the generated ligand molecules and the pocket conformations. For ligand molecules, we evaluate the binding affinity with AutoDock Vina (Eberhardt et al., 2021). There are three modes in AutoDock Vina: "vina_score_only", "vina_minimize", and "vina_dock". We use "vina_minimize" as reported **Vina Score** because this mode performs slight local structure energy minimization without disregarding the generated molecular conformations before estimation. We find an issue with the Vina Score for the complex 6SD9, so we exclude it from the 50 test samples. For each test target, we generate 100 molecules and select the one with the best Vina Score as the final design. For a fair comparison, all methods are

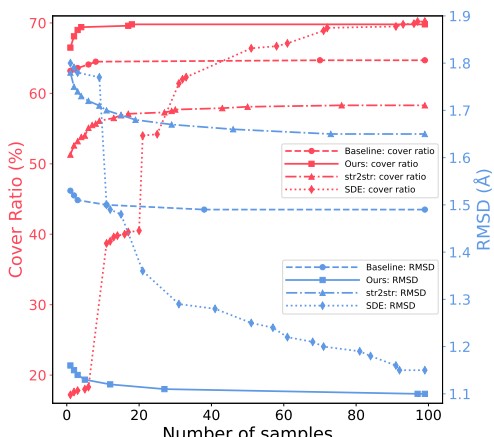

Figure 5: Cover Ratio and minimum RMSD against holo states along the number of samples.

provided with apo pockets as input. We collect the designed ligand molecule of all the test targets and report the mean and standard derivation of Vina Score and the following molecular properties: drug-likeness **QED** (Bickerton et al., 2012), synthesizability **SA** (Ertl & Schuffenhauer, 2009), the octanol-water partition coefficient **logP** (values ranging from -0.4 to 5.6 indicate good drug candidates) (Ghose et al., 1999), number of molecules that obeys Lipinski's rule of five **Lipinski** (Lipinski et al., 2012), the ratio of design molecules that exhibit higher binding affinity than the reference ligand in the test set **High Affi.** (High Affinity). We also report the ratio of generated molecules that are valid in valency and form a single connected graph as **Comp. Rate** (Complete Rate) for the diffusion-based methods. Since autoregressive models filter out the invalid entities during sampling and always form a connected graph, we do not report their Comp. Rate. For pocket conformation, we compute the minimum RMSD between the generated pocket conformation and ground-truth holo conformations for each target and report the average value over all the targets. We compute

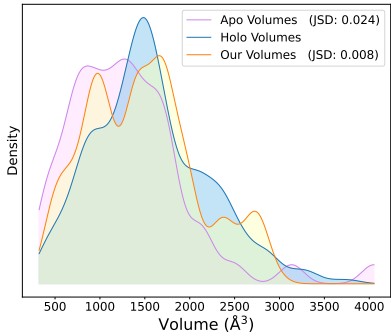

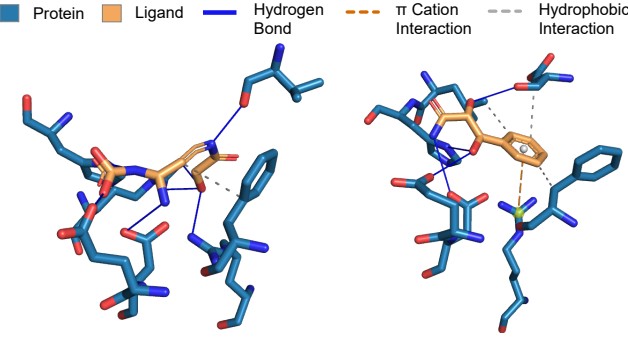

Figure 6: Distribution of apo / holo / our pocket volumes and Jensen-Shannon divergence (JSD) between apo / our and holo volumes.

Figure 7: Visualization of the protein-ligand non-covalent interactions of molecules generated by TargetDiff and apo (left) / pockets refined by our methods (right). Only interacted residues are visualized.

the maximum RMSD between holo conformations for each complex, whose average is 1.42 Å. A ground-truth holo conformation is considered covered when at least one generated pocket conformation has an RMSD less than 1.42 Å against it. We compute the ratio of covered holo conformations for each target and report the average of this value over all the targets as **Cover Ratio**. We also compute the pocket volume using POcket Volume MEasurer 3 (POVME 3) (Wagner et al., 2017) and compare the distribution of volumes between apo/our pockets and ground-truth holo pockets. To further test the generated pocket conformations, for each target, we select the predicted pocket conformation corresponding to the designed ligand molecule with the best Vina Score among 100 candidates as "our pocket". We then use TargetDiff to design ligand molecules for "our pocket", summarize the properties of the designed molecules, and also profile the protein-ligand interactions using Protein-Ligand Interaction Profiler (PLIP) (Salentin et al., 2015).

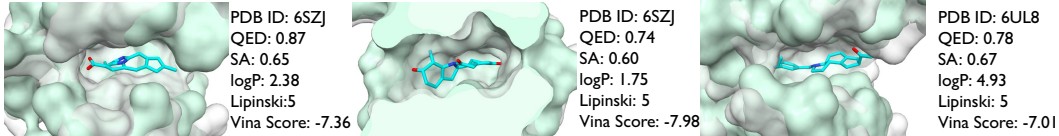

Figure 8: Examples of ligands and holo pockets generated by our model. The green surfaces are the holo pockets and the white surfaces with transparency are the apo pockets.

**Main Results.** We report the properties of designed ligand molecules in Table 1. The molecules designed by our method show high binding affinity and also exhibit satisfactory drug-likeness and synthesizability. The performance of the baselines might be affected by the apo structures, which may not be suitable for designing a binding ligand. Fig. 5 shows the Cover Ratio and minimum RMSD against holo states along the number of samples. DYNAMICFLOW-ODE shows the best performance in RMSD, while DYNAMICFLOW-SDE shows the best performance in Cover Ratio due to the diversity enhanced by SDEs. Str2Str performs even worse than the naive perturbation baseline, possibly because it is designed for whole proteins rather than pockets and does not use sequence information as our methods do. Fig. 6 show the distribution of apo/holo/our pocket volumes. Our pocket volume distribution more closely resembles holo pockets than apo pockets. Table 2 shows that our refined pocket conformation can effectively improve the performance of SBDD methods with rigid pocket inputs. Fig. 4 shows the distribution in number differences of NCIs of apo/our pockets and molecules designed by Targetdiff (see an example in Fig. 7) against those of holo pockets along with the mean of differences in NCI numbers. These results demonstrate that our method effectively transforms apo states into holo-like states. Examples of generated examples are shown in App. G. More ablation studies can be found in Fig. 8 and App. F.

## 5 CONCLUSION

In this work, we introduce a generative modeling approach, DYNAMICFLOW, effectively transforming pockets from apo to holo states and generating ligand molecules simultaneously. Our method uncovers promising ligands and provides enhanced inputs for traditional drug discovery, advancing practical applications in the field. Our research lays the groundwork for integrating protein pocket dynamics into structure-based drug design, with future progress expected.

## ACKNOWLEDGMENTS

This work was supported by the National Key Plan for Scientific Research and Development of China (2023YFC3043300), China's Village Science and Technology City Key Technology funding and Wuxi Research Institute of Applied Technologies, Tsinghua University under Grant 20242001120.

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

# A  DATASETS DETAILS

## A.1  INTRODUCTION OF MISATO DATASET

The limited availability of protein-ligand interaction data has significantly hindered the progress of artificial intelligence in biological domains (Yang et al., 2020). Most of existing datasets either lack molecular dynamics information relevant to protein-ligand interactions (Wang et al., 2005; Liu et al., 2007). The absence of kinetic information for protein-ligand complexes makes it challenging for AI to fully capture and model these biological processes. The development of the MISATO dataset has, to some extent, alleviated the shortage of kinetic data for protein-molecule complexes.

The MISATO dataset (Siebenmorgen et al., 2024) contains the largest molecular dynamics simulation data to date for protein-ligand complexes, along with recalculated QM properties of small molecules. Our work utilizes the complex structures and the associated MD simulation results as the primary data source, with further processing to tailor the data for in-depth modeling.

The development of MISATO dataset is based on the PDBbind database (Wang et al., 2005) which contains around 20,000 complexes structures and experimental binding affinities measurement. A total of 19,443 complexes are collected from PDBBind for further processing. Some important properties during the binding process, such as the binding affinity and the interaction energy are recalculated. Most importantly, the MISATO dataset performs 10ns MD simulations for 16,972 protein-ligand complexes[1], with 8 ns of MD trajectory recorded. These extensive MD results can provide additional kinetic information in the protein-ligand interaction process which is crucial for drug discovery and design research (Vajda et al., 2018).

## A.2  HOLO POCKET CONSTRUCTION

**Holo Pocket Definition.** We exclude the complexes in MISATO where the ligands are oligopeptides, resulting in a total of 12,695 different complexes MD data. To capture different binding pocket conformations from the MD simulation trajectories, we locate residues within a cutoff distance of 7 Å around each ligand and extract them from the 100-frame MD results. We then define the pocket as the union set of all extracted residues from each frame. To select representative holo-state pocket conformations that appear with higher probability in the interaction process, we cluster the 100-frame complexes conformations from the 8 ns MD simulations using GROMACS (Van Der Spoel et al., 2005), with an RMSD threshold of less than 1 Å between the heavy atoms of both the proteins and the ligands. We then retain the top 10 clusters (or fewer if the total number of clusters are smaller than 10) as different holo conformations in our training dataset. The statistics of clustered conformations and pockets are shown in Fig. 9.

Additionally, we refer to the MD trajectory analysis results of the complexes, including the buried solvent-accessible surface area (SASA) (Lee & Richards, 1971), the center-of-mass distance between ligands and receptors, $RMSD_{Ligand}$ (the root-mean-square deviation of the ligand after protein alignment with the native structure), and the molecular mechanics generalized Born surface area (MMGBSA) (Wang et al., 2019). These labels provide a global description of the dynamic properties of the complexes and can be used as evaluation metrics for the binding process. We use the record of $RMSD_{Ligand}$ in the following processing.

**Structural File Alignment.** The PDB files in the MISATO dataset contain unusual three-letter-code representations for residues. This is due to the requirement of the force field calculation, where it needs to consider the special states of amino acid such as protonation and deprotonation for a better simulation accuracy (Case et al., 2021). However, for the training data points in our structure-based drug design model, it is necessary to uniform the representations of residues with the 20 types of three-letter codes for standard amino acids. Consequently, we correct the CYX (Cystine), which denotes the CYS (Cysteine) that forms a disulfide bond with another CYS, and the HIE, which represents the default protonation state of HIS (Histidine) in Amber's pre-processing (Case et al., 2021).

We also examine the atom names within residues and find that some atoms in ASP, GLU, PHE, TRP, TYR occasionally have inconsistent atom naming, which could lead to mismatch in atom-level alignment. Additionally, the Amber adds terminal cappings for the missing residues before MD

---

[1]According to Siebenmorgen et al. (2024), structures from PDBbind were excluded whenever non-standard ligand atoms or inconsistencies in the protein starting structures were encountered, resulting in 16972 complexes for MD simulation.

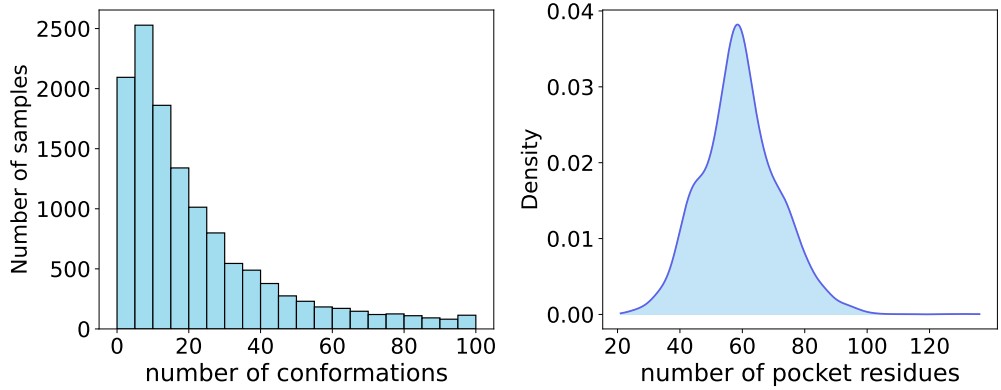

Figure 9: The statistics on the number of conformations (left) and the number of pocket residues (right) in our clustered complexes.

simulations, affecting atom naming in N-terminus and C-terminus residues. These naming issues have been corrected in our processed holo-pocket data. Finally, the residues of all holo proteins are idealized with determined bond lengths and angles according to Engh & Huber (2006).

### A.3 APO POCKET CONSTRUCTION

**Apo Protein Prediction.** In order to obtain the apo-pocket structures as the input of our model, we consider the AlphaFold2-predicted (Jumper et al., 2021) structures as apo-like conformations and extract the corresponding pocket residues to generate apo-like pockets.

One challenge of folding structures directly from current sequences is the lack of chain annotations in the original MISATO files, which causes residues to be indexed continuously from start to finish. This can lead to incorrect folding structures during single-chain folding. Moreover, missing residues in PDB records can further decrease the prediction accuracy of folding models.

Instead, we retrieve AlphaFold2-predicted structures from the AlphaFold Protein Structure Database[2], where proteins are identified using UniProt IDs (Consortium, 2019). We perform ID mapping from Protein Data Bank (PDB)[3] to UniProt using the mapping tools provided by the UniProt API. In total, we successfully map 12,695 PDB IDs to 13,775 UniProt IDs, with 203 PDB IDs failing to find associated UniProt records. Among the successful mappings, 991 entries lack AlphaFold2-predicted structures in the database. Finally, we download AlphaFold2-predicted structures for 2,639 unique proteins and align these structures with the corresponding MISATO holo proteins.

**Apo Pocket Alignment.** To identify the pocket residue indices in apo proteins, we first extract sequences from both the MISATO PDB files and the AlphaFold2-predicted structures, then perform sequence alignment to establish residue index-mapping relationships. We use the Bio.pairwise2 module with the Smith–Waterman algorithm (Smith et al., 1981) for local sequence alignment (Cock et al., 2009). Using the residue index-mapping results, we extract the aligned residues from the apo-protein files to generate what we refer to as apo pockets. Through this workflow, we successfully obtain 7,528 atom-level aligned apo pockets. Alignment failures are primarily due to missing residues, mutations, and aligning sequences for multimers where the pocket consists of residues from different chains. In our curated dataset, the residues forming the pocket all originate from a single chain. Finally, the residues of all holo proteins are idealized with determined bond lengths and angles according to Engh & Huber (2006).

### A.4 LIGAND DATA PROCESSING

The $RMSD_{Ligand}$ is used to evaluate the conformational stability of complexes during MD simulations. To ensure the reliability of MD results for our training data, we filter the clustered holo structures based on the average $RMSD_{Ligand}$ over the 100-frame MD simulation results. A threshold of 3 Å is applied, resulting in a selection of 5,692 holo and apo structures for our training dataset. Additionally, we use RDKit (Landrum et al., 2020) to recalculate the bond connections and bond types in ligands to ensure that all ligands can be successfully loaded and featurized.

---

[2] https://alphafold.ebi.ac.uk
[3] https://www.rcsb.org/

The protein-ligand complex conformations in our training dataset count for 46,235. The distribution of $RMSD_{Ligand}$ and the number of remaining complexes after each key processing step are illustrated in Fig. 10.

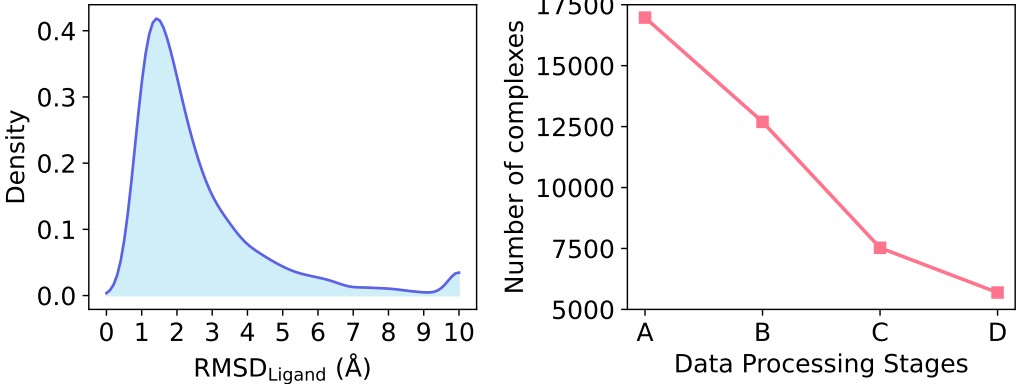

Figure 10: Left: Statistics of $RMSD_{Ligand}$ for complexes in MISATO; Right: Statistics on dataset size after each processing step. (A-B) Filtering out complexes of where ligands are peptides; (B-C): Sequence alignment and apo-pocket extraction; (C-D): Filtering conformation ensembles with $RMSD_{Ligand}$ smaller than 3 Å.

## A.5 VISUALIZATIONS OF DATASET

We provide visualizations of apo pockets and holo conformations from our processed dataset in Fig. 13. In each row, the left figure shows the apo pocket and the right two figures are different holo pockets with the same ligand. We also visualize the main chain and side chains of residues surrounding the ligand. Additionally, we evaluate the ligands in our dataset with QED, SA, logP, Lipinski and their binding affinity with apo and holo pockets using vina_score and vina_min. The results are presented in Fig. 11 and Fig. 12.

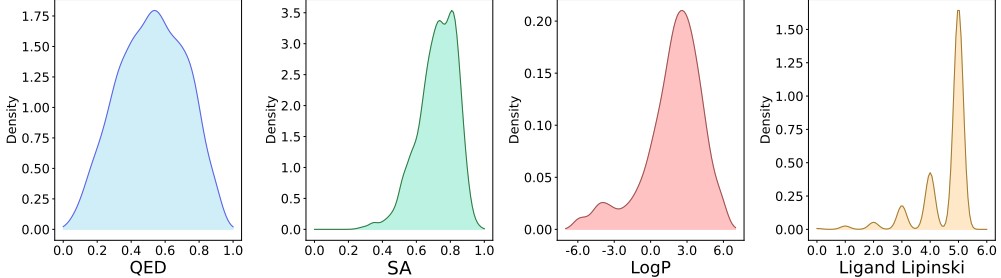

Figure 11: The distribution of molecular properties (QED, SA, logP, and Lipinski) for ligands in our dataset.

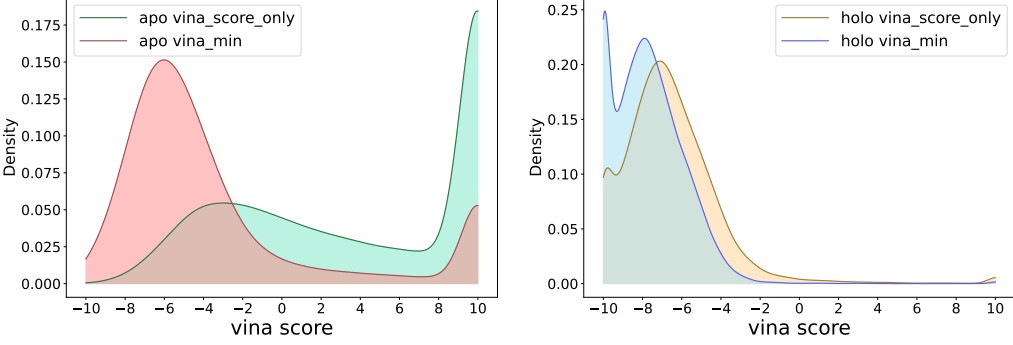

Figure 12: The distribution of vina_score_only and vina_min of apo/holo complexes in our dataset.

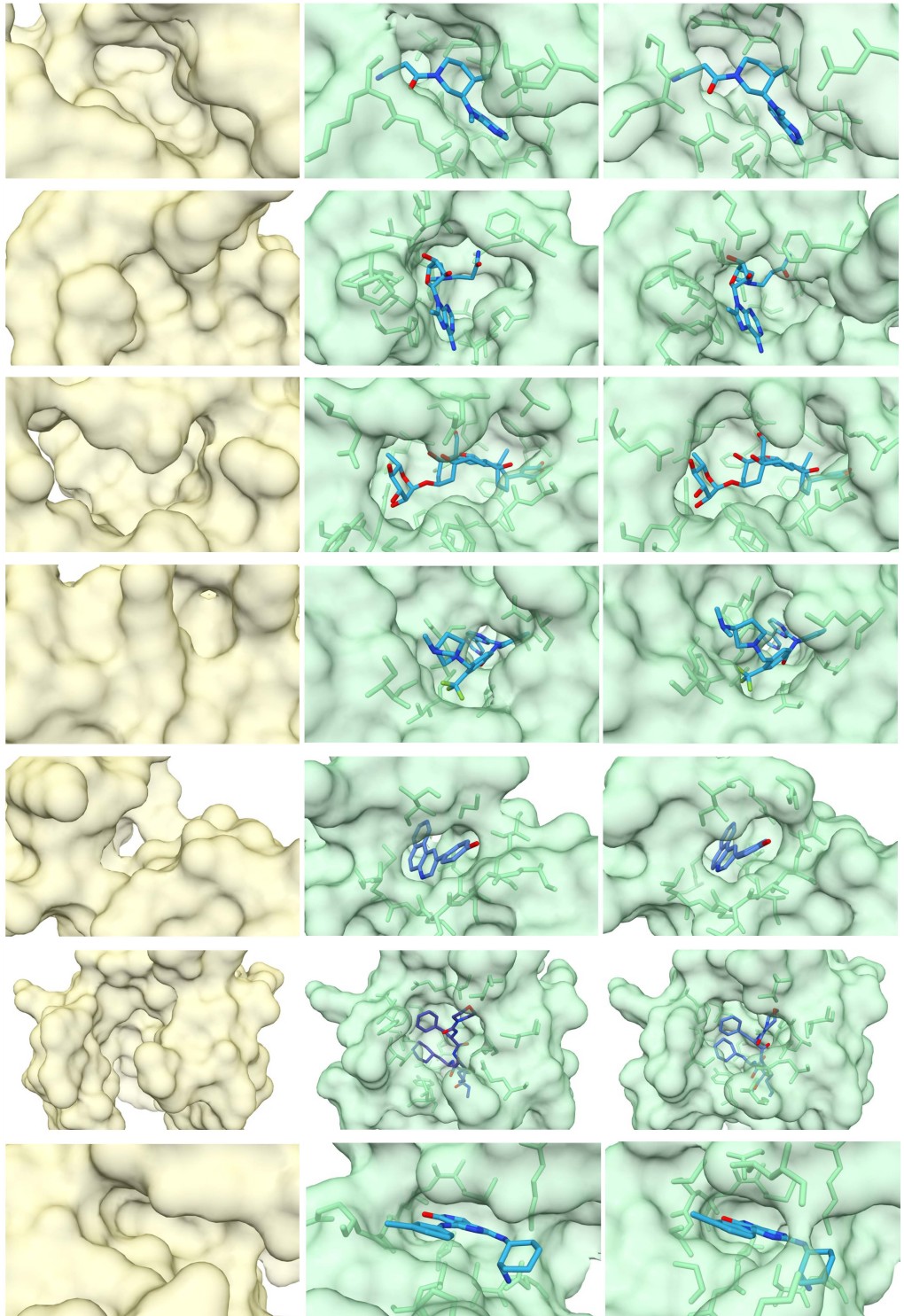

Figure 13: Examples of apo pockets and holo complexes from our processed dataset. The left yellow regions represent the apo pockets in the form of protein surfaces. The right two green regions show different holo conformations of the same proteins and ligands. The PDB IDs for these examples are 3FUP, 3DZ4, 3A3Y, 2E2B, 1PMU, 1FQ8, and 5TT7.

## B    ALL ATOM COORDINATES COMPUTING

We adopt a coordinates computing and updating method similar to those employed in previous works (Campbell et al., 2024; Jumper et al., 2021; Yim et al., 2023b). The residue-specific frames are first constructed to describe the idealized atom positions in every type of amino acid. Then we will predict the backbone and side-chain frames update and map the updated frames back to atom coordinates. We will explain the details of this process in the following paragraphs.

**Backbone Parameterization.** To simplify the expressions, we specify that all subsequent expressions refer to the residue at index $i$. We begin by constructing the frame centered at $C_\alpha$ for every residue using their position of $C_\alpha$, C and N, represented by bold letters. The residues are idealized with determined bond lengths and angles according to Engh & Huber (2006). Based on this setting, the rigid frame is built with the Gram-Schmidt process:

$$\mathbf{v}_1 = \mathbf{C} - \mathbf{C}_\alpha, \qquad \mathbf{v}_2 = \mathbf{N} - \mathbf{C}_\alpha, \qquad \mathbf{e}_1 = \mathbf{v}_1/\|\mathbf{v}_1\|,$$
$$\mathbf{u}_2 = \mathbf{v}_2 - \mathbf{e}_1(\mathbf{e}_1^\top \mathbf{v}_2), \qquad \mathbf{e}_2 = \mathbf{u}_2/\|\mathbf{u}_2\|, \qquad \mathbf{e}_3 = \mathbf{e}_1 \times \mathbf{e}_2. \tag{27}$$

Then we can generate backbone amino acid frame $T_0 = (\mathbf{t}_0, \mathbf{r}_0)$ with $\mathbf{t}_0 = \mathbf{C}_\alpha = \mathbf{0}$ and $\mathbf{r}_0 = \text{concat}(\mathbf{e}_1, \mathbf{e}_2, \mathbf{e}_3) = \mathbf{I}$.

**Backbone Frame Update.** According to Jumper et al. (2021), the rotation and translation of backbone are expressed using a predicted quaternion $(a, b, c, d)$ and vector $\mathbf{t}$. The rotation matrix $\mathbf{r}$ is calculated from the predicted quaternion as follows:

$$\mathbf{r} = \begin{pmatrix} a^2 + b^2 - c^2 - d^2 & 2bc - 2ad & 2bd + 2ac \\ 2bc + 2ad & a^2 - b^2 + c^2 - d^2 & 2cd - 2ab \\ 2bd - 2ac & 2cd + 2ab & a^2 - b^2 - c^2 + d^2 \end{pmatrix}. \tag{28}$$

Note that the quaternion need normalization before the calculation of $\mathbf{r}$. The backbone frame update can thus be expressed as $T_{bb} = (\mathbf{t}, \mathbf{r})$.

The position of the O atom in the backbone is constrained to a circle that rotates around the x-axis (defined by $C_\alpha$-C bond) with the C-O bond. We predict the torsion angle of the backbone $\chi_0 = [\chi_{01}, \chi_{02}] \in \text{SO}(2)$ that determines the position of the O atom and calculate the rotation with:

$$\mathbf{r}_x^{\chi_0} = \begin{pmatrix} 1 & 0 & 0 \\ 0 & \chi_{01} & -\chi_{02} \\ 0 & \chi_{02} & \chi_{01} \end{pmatrix}, \tag{29}$$

where $\|(\chi_{01})^2 + (\chi_{02})^2\| = 1$, the C-O bond length which defines the translation $\mathbf{t}_{\chi_0}$ is specified at 1.23 Å. Using $T_{\chi_0} = (\mathbf{t}_{\chi_0}, \mathbf{r}^{\chi_0})$ with backbone frame update, we can update the position of the O atom in backbone.

**FrameToAtom Mapping.** The updating of full atom positions involves the backbone rotation and translation with side-chain torsion $\alpha = (\chi_1, \chi_2, \chi_3, \chi_4)$. For each torsion angle, an additional side-chain frame is built to ensure that the torsion occurs around the x-axis within the current frame (Jumper et al., 2021). The definitions of the side-chain torsion angles for the 20 types of amino acids are shown in Table 3. The side-chain frames are updated hierarchically as follows:

$$T_1 = T_{bb} \circ T_{(\chi_1 \to bb)}^{\mathbf{a}} \circ \mathbf{r}_x^{\chi_1},$$
$$T_2 = T_1 \circ T_{(\chi_2 \to \chi_1)}^{\mathbf{a}} \circ \mathbf{r}_x^{\chi_2},$$
$$T_3 = T_2 \circ T_{(\chi_3 \to \chi_2)}^{\mathbf{a}} \circ \mathbf{r}_x^{\chi_3},$$
$$T_4 = T_3 \circ T_{(\chi_4 \to \chi_3)}^{\mathbf{a}} \circ \mathbf{r}_x^{\chi_4},$$
$$\mathbf{r}_x^{\alpha} = \begin{pmatrix} 1 & 0 & 0 \\ 0 & \alpha_1 & -\alpha_2 \\ 0 & \alpha_2 & \alpha_1 \end{pmatrix}. \tag{30}$$

Here, $T_{(\chi_1 \to bb)}^{\mathbf{a}}$ denotes the transformation from the $\chi_1$ frame to the backbone frame for an idealized amino acid of type $\mathbf{a}$. The same hierarchical updating process is applied from $\chi_1$ to $\chi_4$. $\mathbf{r}_x^{\alpha}$ represents the rotation of the four types of torsion angles around $x$-axis where $\|(\alpha_1)^2 + (\alpha_2)^2\| = 1$. Using the

Table 3: Definition of the torsion angles ($\chi_1$, $\chi_2$, $\chi_3$, $\chi_4$) and their corresponding rigid groups for the 20 types of residues. We frame the torsion angles that are $\pi$-symmetric and list the required torsion angles to fix the side-chain atom positions.

| Residue Type | $\chi_1$ | $\chi_2$ | $\chi_3$ | $\chi_4$ | Atom position fixed by |
|---|---|---|---|---|---|
| ALA (A) | - | - | - | - | - |
| ARG (R) | $N, C^\alpha, C^\beta, C^\gamma$ | $C^\alpha, C^\beta, C^\gamma, C^\delta$ | $C^\beta, C^\gamma, C^\delta, N^\epsilon$ | $C^\gamma, C^\delta, N^\epsilon, C^\zeta$ | $\chi_1, \chi_2, \chi_3, \chi_4$ |
| ASN (N) | $N, C^\alpha, C^\beta, C^\gamma$ | $C^\alpha, C^\beta, C^\gamma, O^{\delta 1}$ | - | - | $\chi_1, \chi_2$ |
| ASP (D) | $N, C^\alpha, C^\beta, C^\gamma$ | $\boxed{C^\alpha, C^\beta, C^\gamma, O^{\delta 1}}$ | - | - | $\chi_1, \chi_2$ |
| CYS (C) | $N, C^\alpha, C^\beta, S^\gamma$ | - | - | - | $\chi_1$ |
| GLN (Q) | $N, C^\alpha, C^\beta, C^\gamma$ | $C^\alpha, C^\beta, C^\gamma, C^\delta$ | $C^\beta, C^\gamma, C^\delta, O^{\epsilon 1}$ | - | $\chi_1, \chi_2, \chi_3$ |
| GLU (E) | $N, C^\alpha, C^\beta, C^\gamma$ | $C^\alpha, C^\beta, C^\gamma, C^\delta$ | $\boxed{C^\beta, C^\gamma, C^\delta, O^{\epsilon 1}}$ | - | $\chi_1, \chi_2, \chi_3$ |
| GLY (G) | - | - | - | - | - |
| HIS (H) | $N, C^\alpha, C^\beta, C^\gamma$ | $C^\alpha, C^\beta, C^\gamma, N^{\delta 1}$ | - | - | $\chi_1, \chi_2$ |
| ILE (I) | $N, C^\alpha, C^\beta, C^{\gamma 1}$ | $C^\alpha, C^\beta, C^{\gamma 1}, C^{\delta 1}$ | - | - | $\chi_1, \chi_2$ |
| LEU (L) | $N, C^\alpha, C^\beta, C^\gamma$ | $C^\alpha, C^\beta, C^\gamma, C^{\delta 1}$ | - | - | $\chi_1, \chi_2$ |
| LYS (K) | $N, C^\alpha, C^\beta, C^\gamma$ | $C^\alpha, C^\beta, C^\gamma, C^\delta$ | $C^\beta, C^\gamma, C^\delta, C^\epsilon$ | $C^\gamma, C^\delta, C^\epsilon, N^\zeta$ | $\chi_1, \chi_2, \chi_3, \chi_4$ |
| MET (M) | $N, C^\alpha, C^\beta, C^\gamma$ | $C^\alpha, C^\beta, C^\gamma, S^\delta$ | $C^\beta, C^\gamma, S^\delta, C^\epsilon$ | - | $\chi_1, \chi_2, \chi_3$ |
| PHE (F) | $N, C^\alpha, C^\beta, C^\gamma$ | $\boxed{C^\alpha, C^\beta, C^\gamma, C^{\delta 1}}$ | - | - | $\chi_1, \chi_2$ |
| PRO (P) | $N, C^\alpha, C^\beta, C^\gamma$ | $C^\alpha, C^\beta, C^\gamma, C^\delta$ | - | - | - |
| SER (S) | $N, C^\alpha, C^\beta, O^\gamma$ | - | - | - | $\chi_1$ |
| THR (T) | $N, C^\alpha, C^\beta, O^{\gamma 1}$ | - | - | - | $\chi_1$ |
| TRP (W) | $N, C^\alpha, C^\beta, C^\gamma$ | $C^\alpha, C^\beta, C^\gamma, C^{\delta 1}$ | - | - | $\chi_1, \chi_2$ |
| TYR (Y) | $N, C^\alpha, C^\beta, C^\gamma$ | $\boxed{C^\alpha, C^\beta, C^\gamma, C^{\delta 1}}$ | - | - | $\chi_1, \chi_2$ |
| VAL (V) | $N, C^\alpha, C^\beta, C^{\gamma 1}$ | - | - | - | $\chi_1$ |

above frame transformations and updating rules, we can mapping the idealized atom positions from the frames to coordinates, also referred to as FrameToAtom in the main paragraph:

$$\hat{x}_{\mathbf{v}}^{\mathbf{a}} = \text{concat}_{f, \mathbf{v}'} \left( \left( T_f \circ x_{f, \mathbf{v}'}^{\mathbf{a}} \right) \right), \tag{31}$$

where $\hat{x}_{\mathbf{v}}^{\mathbf{a}}$ is the updated atom position, $x_{f, \mathbf{v}'}^{\mathbf{a}}$ represents the idealized position of atom $\mathbf{v}$ in residue with amino acid type $\mathbf{a}$ under frame $f$ defined by different rotations and translations in Eqs. (28) to (30).

## C   DEFINITION OF TORSION ANGLES

In Table 3, we present the compositions of all the side chain torsion angles in the 20 types of amino acids that are used to compute atomic coordinates. The side-chain torsion angles that show $\pi$-symmetric are highlighted with black boxes.

## D   ILLUSTRATION OF THE OVERALL WORKFLOW

We illustrate the overall workflow in Fig. 14 to enhance understanding, particularly of how different components of the multi-scale model relate to the training loss.

## E   NOTE ON EVALUATION METRICS

From a distribution learning standpoint, generated ligands with more similar statistics (e.g., mean and median of QED, SA, Vina Score) to reference ligands indicate a superior model. In our work, we followed conventions from other studies (Guan et al., 2023) by using "↑" or "↓" to denote preferences in drug design. From the distribution learning perspective, our methods outperform others across nearly all metrics, as they most closely approximate the reference molecules in terms of property statistics.

## F   ABLATION STUDIES

**Effect of Interaction Loss.** We perform ablation studies to verify the effect of interaction loss that is introduced in Section 3.2. The related results are reported in Table 4. The results demonstrate that the proposed interaction loss can effectively improve the performance of both ODE and SDE versions of DYNAMICFLOW, especially the binding affinity.

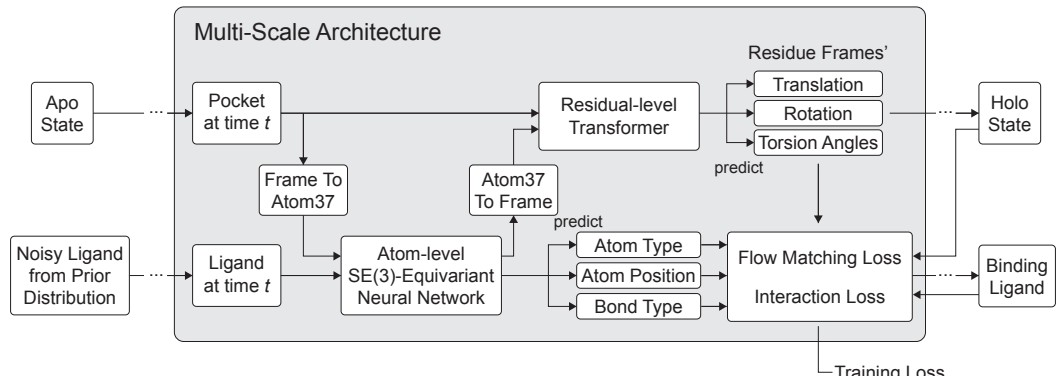

Figure 14: The illustration of the overall workflow. The model inputs are the protein-ligand complexes at time $t$. (Notably, the apo state and noisy ligand sampled from the prior distribution represent the protein-ligand complex at time 0, while the holo state and its binding ligand correspond to the complex at time 1. During training, the complex at time $t$ is interpolated between the samples at times 0 and 1. During sampling, the complex at time $t$ is derived by solving the ODE/SDE defined by the flow model.) The pocket and ligand are first input into the atom-level SE(3)-equivariant neural network. The output features corresponding to the ligand are used to predict the ligand's atom type, atom position, and bond type. Meanwhile, the output features related to the pocket, along with the original residue-level features of the pocket (such as residue type), are input into the residue-level transformer to predict residue frames' translation, rotation, and torsion angles. Ultimately, all of these predictions contribute to computing the training loss.

Table 4: Summary of properties of reference molecules and molecules generated by our model and the variants without interaction loss. ($\uparrow$) / ($\downarrow$) denotes a larger / smaller number is better.

| | Vina Score ($\downarrow$) | QED ($\uparrow$) | SA ($\uparrow$) | Linpiski ($\uparrow$) | logP | High Affi. ($\uparrow$) | Comp. Rate ($\uparrow$) |
|---|---|---|---|---|---|---|---|
| DYNAMICFLOW-ODE | -7.28±1.98 | 0.53±0.20 | 0.61±0.14 | 3.13±2.62 | 4.45±0.81 | 51.0% | 70.2% |
| w/o interaction loss | -6.76±1.39 | 0.54±0.22 | 0.60±0.15 | 3.20±2.48 | 4.36±0.93 | 37.4% | 72.2% |
| DYNAMICFLOW-SDE | -7.65±1.59 | 0.53±0.15 | 0.53±0.17 | 4.34±2.58 | 4.25±0.90 | 52.5% | 73.6% |
| w/o interaction loss | -7.00±1.15 | 0.48±0.21 | 0.56±0.16 | 1.35±3.54 | 4.06±1.48 | 43.4% | 78.0% |

**Effect of Multi-Scale Model Architecture.** We conduct ablation studies to evaluate the impact of different architectural components on model performance. We implement a baseline denoted as "w/o residue-level Transformer", which uses only atom-level SE(3)-equivariant geometrical message-passing layers. In this setup, atom-level output features are aggregated into residue-level features without employing a residue-level Transformer for further extraction, and these aggregated features are used to predict the residue frames' translation, rotation, and torsion angles.

Additionally, we develop a baseline referred to as "w/o atom-level EGNN", which transforms the atom-level protein-ligand complex graph into a heterogeneous graph, where each node represents either a residue (with C-alpha coordinates, rotation vectors, and torsion angles as input features) or a ligand atom. In this variant, since we do not explicitly reconstruct the full atom representation of the pocket, the atom interaction loss is not applied.

Table 5: Summary of properties of reference molecules and molecules generated by our model and the variants with different model architectures. ($\uparrow$) / ($\downarrow$) denotes a larger / smaller number is better.

| | Vina Score ($\downarrow$) | QED ($\uparrow$) | SA ($\uparrow$) |
|---|---|---|---|
| DYNAMICFLOW-ODE | -7.28 ± 1.98 | 0.53 ± 0.20 | 0.61 ± 0.14 |
| w/o interaction loss | -6.76 ± 1.39 | 0.54 ± 0.22 | 0.60 ± 0.15 |
| w/o residue-level Transformer | -6.23 ± 1.68 | 0.53 ± 0.22 | 0.59 ± 0.14 |
| w/o atom-level EGNN | -6.02 ± 1.63 | 0.54 ± 0.19 | 0.64 ± 0.13 |
| DYNAMICFLOW-SDE | -7.65 ± 1.59 | 0.53 ± 0.15 | 0.53 ± 0.17 |
| w/o interaction loss | -7.00 ± 1.15 | 0.48 ± 0.21 | 0.56 ± 0.16 |
| w/o residue-level Transformer | -6.50 ± 1.22 | 0.52 ± 0.16 | 0.56 ± 0.14 |
| w/o atom-level EGNN | -6.13 ± 1.31 | 0.49 ± 0.19 | 0.60 ± 0.16 |

The results are shown in Table 5. The results indicate that our proposed architecture significantly enhances binding affinity and is vital for effectively modeling protein-ligand interactions and protein dynamics.

Both variants ("w/o residue-level Transformer" and "w/o atom-level EGNN") are more computationally efficient due to their reduced model sizes. However, despite using both residue-level and atom-level models, our method maintains acceptable inference speed because our flow model can generate high-quality ligand molecules in fewer steps.

# G   VISUALIZATION OF GENERATED SAMPLES

**Visualization of Generated Complexes.** Here, we list more examples of generated ligands and holo-like pockets in Fig. 15. The ligands demonstrate promising binding affinity, as indicated by their Vina Scores.

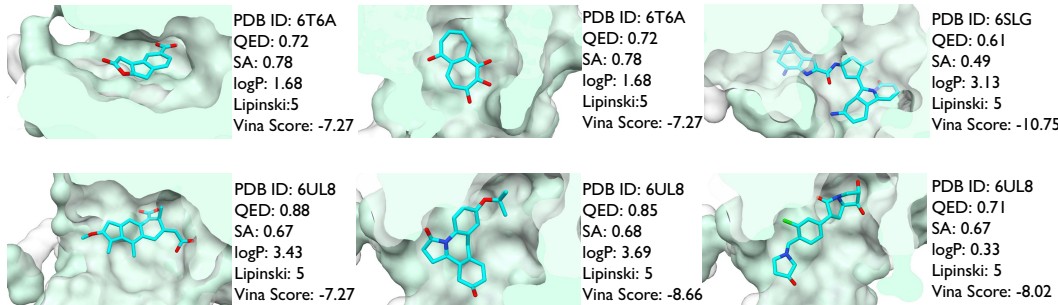

Figure 15: Examples of our generated ligands and corresponding holo pockets. The green surfaces are the holo pockets and the white surfaces with transparency are the apo pockets.

**Visualization of Sampling Trajectories.** We sample from the generation trajectories and illustrate the ligand generation process along with the conformational changes of the pocket in Fig. 16. The reference apo pockets are from 6SZJ (top) and 6UL8 (bottom). From left to right, the ligands are reconstructed from random noise, represented by the nodes. The apo pockets also undergo conformational changes, transitioning from the apo state to the holo state in response to the generated ligands.

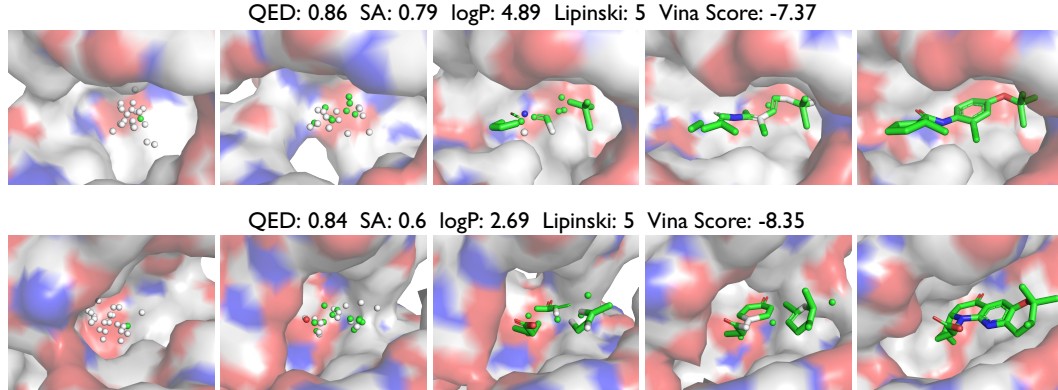

Figure 16: Examples of our generation flow. The ligand is initialized at the CoM of the apo pocket in terms of random noise. The white node corresponds to the mask token of atom type.

# H   EVALUATION ON POCKET VOLUME

We use POVME3 (Wagner et al., 2017) to estimate the volumes of generated pockets and make comparison with the volumes of apo and holo pockets. POVME3 is a receptor-centric method for pocket analysis. It defines pocket volume by occupying the surrounding region of the receptor with equidistant points. Points that are near or outside of receptor atoms are removed and the volume is calculated based on the remaining points. Following this approach, we define the pocket region as the spherical area with a 15 Å radius centered at the pocket's geometric center, excluding points within the van der Waals radius (plus the hydrogen atom radius) of receptor atoms. Compared with the apo pockets, our (randomly selected) generated pockets are more similar to the reference holo

states with smaller volume differences. The results are shown in Table 6. We also show an example of the volume occupation of our generated pocket and the corresponding apo/holo pocket in Fig. 17.

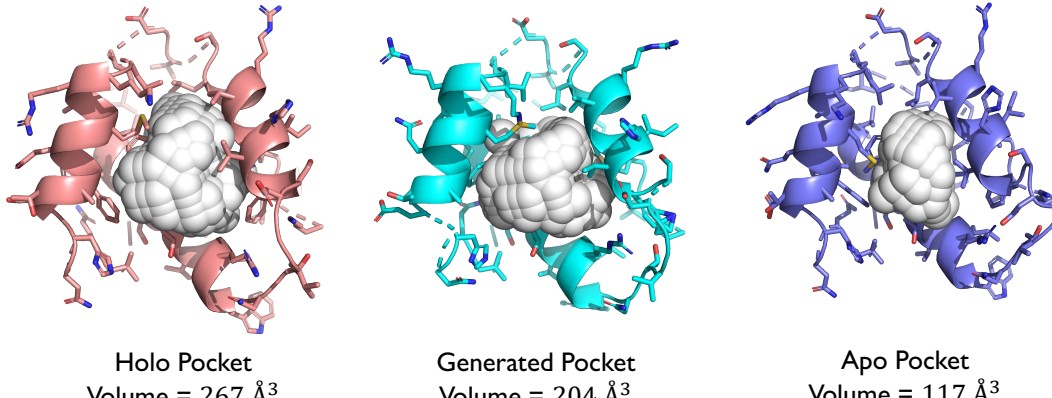

| Holo Pocket | Generated Pocket | Apo Pocket |
| Volume = 267 Å³ | Volume = 204 Å³ | Volume = 117 Å³ |

Figure 17: Example of volume calculation from POVME3 using a 6 Åradius. We select the apo (purple) and holo (red) pocket from PDB file 6UDI, alongside the generated pocket (cyan). Compared to the apo pocket, our generated pocket more resembles the real holo structure in terms of pocket volume.

Table 6: Volume difference (Å³) between apo/our pockets and holo pockets.

|  | Avg. ± Std. | Med. |
| --- | --- | --- |
| Apo states | 83.84 ± 61.20 | 71.20 |
| Our Pocket (DYNAMICFLOW-ODE) | **50.08 ± 35.05** | **41.75** |
| Our Pocket (DYNAMICFLOW-SDE) | 68.56 ± 55.51 | 59.20 |

## I  BINDING AFFINITY EVALUATION CONSIDERING PROTEIN DYNAMICS

Incorporating molecular dynamics in evaluation is also important. However, relying on MD trajectory-based methods such as MMGBSA or MMPBSA (Wang et al., 2019) can be highly cumbersome. These approaches employ different solvation models and require extensive computational resources, with MD simulations often taking months to complete for thousands of systems. Therefore, as an alternative, we opt for a deep-learning-based approach for flexible docking and scoring to achieve reliable and scalable evaluation.

Specifically, for each generated ligand designed by the baselines and our methods, we employ DynamicBind (Lu et al., 2024b), a geometric deep generative model tailored for "dynamic docking", to generate 10 protein-ligand complex structures. DynamicBind also includes a model that predicts an "affinity" score, which estimates the negative logarithm of the binding affinity in concentration units. We then calculate the weighted average of these predicted binding affinities to derive the final "affinity" score, where a higher "affinity" score indicates better binding potential.

For each target, we assess the affinity of a randomly selected generated ligand (single), the highest affinity among 10 generated ligands (Best over 10), the best affinity over all 100 generated ligands (Best over all), and the average affinity over all 100 generated ligands (Avg. over all). We report the mean, standard deviation, and median of these affinities across 50 targets. The results are summarized in Table 7.

## J  INFERENCE TIME

We benchmark the inference time of baselines and our methods for generating 10 ligand molecules given the same pocket on 1 Tesla V100-SXM2-32GB. The default number of function evaluations (NFE) is 1000 for TargetDiff and IPDiff and 100 for our method.

As Table 8 shows, our methods are capable of generating high-quality ligands while simultaneously modeling protein dynamics at a fast speed, demonstrating a significant advantage in computational efficiency.

Table 7: Binding affinity evaluation based on DynamicBind.

|  | Single | | Best over 10 | | Best over All | | Avg. over All | |
| --- | --- | --- | --- | --- | --- | --- | --- | --- |
|  | Avg. ± Std. | Med. | Avg. ± Std. | Med. | Avg. ± Std. | Med. | Avg. ± Std. | Med. |
| Pocket2Mol | 3.64 ± 1.26 | 3.31 | 4.90 ± 1.15 | 4.81 | 5.70 ± 1.22 | 5.68 | 3.74 ± 0.92 | 3.41 |
| TargetDiff | 6.00 ± 1.14 | 6.19 | 7.30 ± 0.70 | 7.46 | 7.81 ± 0.71 | 7.91 | 6.14 ± 0.82 | 6.29 |
| TargetDiff* | 6.19 ± 0.97 | 6.38 | 7.16 ± 0.94 | 7.54 | 7.64 ± 0.73 | 7.79 | 6.08 ± 0.92 | 6.21 |
| IPDiff | 6.15 ± 1.14 | 6.45 | 7.05 ± 0.79 | 7.18 | 7.68 ± 0.90 | 7.82 | 6.05 ± 0.84 | 6.19 |
| IPDiff* | 5.96 ± 1.31 | 5.83 | 7.10 ± 1.09 | 7.14 | 7.63 ± 0.97 | 7.72 | 6.10 ± 1.10 | 6.12 |
| DYNAMICFLOW-ODE | **6.46 ± 1.00** | **6.69** | 7.40 ± 0.94 | 7.62 | 7.91 ± 0.90 | 8.07 | 6.37 ± 0.97 | **6.38** |
| DYNAMICFLOW-SDE | 6.21 ± 1.19 | 6.09 | **7.53 ± 0.86** | **7.67** | **7.95 ± 0.83** | **8.12** | **6.39 ± 0.87** | 6.37 |

Table 8: Inference time of baselines and our methods for generating 10 ligand molecules.

|  | Time (s) | Default NFE |
| --- | --- | --- |
| Pocket2Mol | 980 | N/A |
| TargetDiff | 156 | 1000 |
| TargetDiff* | 154 | 1000 |
| IPDiff | 334 | 1000 |
| IPDiff* | 343 | 1000 |
| DYNAMICFLOW-ODE | 35 | 100 |
| DYNAMICFLOW-SDE | 36 | 100 |

## K IMPLEMENTATION DETAILS

In this section, we provided details of the implementation of our methods.

**Featurization.** Protein and ligand embeddings (as shown in Fig. 3) are derived from the encodings of protein atom features and ligand atom and bond features, respectively, through an embedding layer (i.e., learnable linear transformation). The protein atom feature contains its atom37 representation and residue type. Atom37 is an all-atom representation of proteins where each heavy atom corresponds to a given position in a 37-dimensional array. This mapping is non amino acid specific, but each slot corresponds to an atom of a given name. Note that atom37 is widely used in protein modeling (Jumper et al., 2021). We concatenate the one-hot encodings of these two features (whose dimensions are 37 and 20, respectively) to derive the protein atom encodings (whose dimension is 57). We use the one-hot encoding of the atom type as the ligand atom encoding. We only consider explicitly modeling "C, N, O, F, P, S, Cl, Br" in ligands, so the dimension is 8. For ligand bond types, we consider "non-bond, single, double, triple, aromatic", so the dimension of ligand bond encoding is 5. The protein and ligand atom features are used as the initial node features in the complex graph. And ligand atom and bond features are used as the initial node and edge features, respectively, in the ligand graph. The above encodings are common in modeling protein and small molecules.

**Hyperparameters.** There are 7 individual losses: 4 continuous flow matching losses for residue frames' translation (Eq. (5)), rotation (Eq. (7)), torsion angles (Eq. (8)) and ligand atom position (same as Eq. (5)), 2 discrete flow matching losses for ligand atom and bond types (Eq. (14)), and interaction loss (Eq. (18)). There are first averaged across all residues or atoms in a training sample and then simply weighted summed with weights: 2.0, 1.0, 1.0, 4.0, 1.0, 1.0, 0.5.

We use AdamW (Loshchilov, 2017) as the optimizer with learning rate 0.0002, beta1 0.95, and beta2 0.999. $\gamma$ controls the stochasticity of the stochastic flow (see Eqs. (19) to (21)). We use 2.0, 0.005, 1.0, 2.0 as the values of $\gamma$ for residue frames' translation, rotation, torsion angles, and ligand atom positions.

**Model Architecture.** Our model consists of an atom-level SE(3)-equivariant graph neural network and a residue-level Transformers. The number of total parameters is 15.9 M. The total estimated model parameter size is 63.401 MB. We include more details about the model architecture in Table 9.

## L EXTENDED RELATED WORKS

In this section, we discuss more related works in addition to those mentioned in Section 2.

Table 9: Details of our model architecture.

|  | Layer name | Number of layers |
|---|---|---|
| Atom-level Model | Protein atom embedding layer | 1 |
|  | Ligand atom embedding layer | 1 |
|  | Ligand bond embedding layer | 1 |
|  | Time embedding layer | 1 |
|  | EGNN block | 6 |
|  | Ligand atom type prediction head | 1 |
|  | Ligand bond type prediction head | 1 |
| Residue-level Model | Protein residue embedding layer | 1 |
|  | Time embedding layer | 1 |
|  | Transformer block with IPA | 4 |
|  | Torsion angle prediction head | 1 |

There are several research efforts that also focus on **structure-based drug design with rigid pockets**. We discuss these works as follows: Ragoza et al. (2022) applied a variational autoencoder to generate 3D molecules within atomic density grids. Guan et al. (2024) introduced decomposed priors and validity guidance to improve the quality of ligand molecules generated by diffusion models. Zhang & Liu (2023) enhanced molecule generation through global interactions between subpocket prototypes and molecular motifs. Zhang et al. (2023a) promoted ligand molecule generation based on the principle of parallel multiscale modeling. Lin et al. (2023) employed diffusion models to generate ligand molecules by denoising the translation and rotation of 3D motifs. Qu et al. (2024) introduced an SBDD model that operates in the continuous parameter space, equipped with a noise-reduced sampling strategy. Huang et al. (2024c) proposed to adaptively extract essential parts of binding sites to enhance the quality of ligand molecules generated by diffusion models. Huang et al. (2024d) leveraged a curated set of ligand references, i.e., those with desired properties such as high binding affinity, to steer the diffusion model towards synthesizing ligands that satisfy design criteria. Zhou et al. (2024a) integrated conditional diffusion models with iterative optimization to optimize properties of generated molecules. Lee et al. (2024) proposed to simultaneously denoise non-covalent interaction types of protein-ligand edges along with a 3D graph of a ligand molecule. Zhou et al. (2024b) and Cheng et al. (2024) proposed to fine-tune diffusion models by policy gradient (Sutton et al., 1999; Lillicrap et al., 2015) and direct preference optimization (Rafailov et al., 2023), respectively, to generate ligand molecules with desired properties. Huang et al. (2024a) represents a standard SBDD method with rigid-pocket input. Although molecular dynamics were mentioned in this work, they refer to dynamics induced by the forward process of the diffusion model. There are other settings for structure-based drug design. For instance, Zhou et al. (2025) proposed to reprogram single-target diffusion models to design dual-target ligands in a zero-shot manner.

A recent work (Schneuing et al., 2024) considers protein flexibility in terms of side-chain torsions in SBDD. Our work differs in that our method models the dynamics of both the protein backbone structure and side-chain torsion, and utilizes rigorous discrete flow matching for modeling atom and bond types of ligand molecules. Importantly, our model employs an apo-holo transformation paradigm, offering a biologically informative prior for the flow model. In contrast, Schneuing et al. (2024) focuses on denoising side-chain torsions from a non-informative prior derived from a static protein-ligand complex dataset.

Another recent research work, FlexSBDD (Zhang et al., 2024b), and our work both integrate protein flexibility or dynamics into SBDD but present differences in various aspects. We developed our methods independently from FlexSBDD. The key distinctions between our work and FlexSBDD include:

- **Motivation:** FlexSBDD primarily seeks to incorporate protein flexibility into SBDD for optimizing complex structures and ligands. However, it overlooks the role of thermodynamic fluctuations that govern protein flexibility and conformational shifts, leading to diverse conformations with different ligands. In contrast, our work delves into the physics underlying these dynamics. We illustrate this by examining the DFG-in and DFG-out states of Abl kinase (see Fig. 1), emphasizing our motivation to integrate comprehensive protein dynamics into SBDD, beyond merely addressing flexibility.

- **Data:** FlexSBDD derives most of its apo data by augmenting holo data through relaxation/sidechain repacking. In contrast, we use AlphaFold2 to predict our apo data, poten-

tially resulting in greater conformational changes. Additionally, our holo states are diverse, providing multiple states for each protein-ligand pair through molecular dynamics simulations, enabling a more thorough exploration of pocket conformational changes. This aligns with our motivation.

- **Methodology:**
  - **Protein Modeling:** FlexSBDD employs a residue-level model for the protein pocket, whereas we utilize both residue-level and atom-level models simultaneously, leveraging atom37 mapping. This approach allows us to more precisely capture protein-ligand interactions at the atomic level, enhancing the accuracy and detail of our modeling.
  - **Ligand Modeling:** On the ligand side, we construct flow models for atom positions, atom types, and bond types simultaneously in an end-to-end manner. In contrast, FlexSBDD does not incorporate bond modeling within its flow model, instead generating bonds through empirical post-processing rules. Our approach allows for more integrated and cohesive modeling of ligand structures.
  - **Discrete Variable Modeling:** FlexSBDD uses continuous vectors to represent discrete variables (i.e., atom types) and utilizes standard flow matching for continuous variables, employing "norm" for self-normalization to mimic probabilities. This introduces a lack of rigor due to the inference gap created by "norm". Conversely, we apply rigorous discrete flow matching using continuous-time Markov chains (CTMC) to model both atom and bond types, ensuring a more precise and theoretically robust representation. For detailed mathematical insights, see Section 3.1 and Section 3.2.
  - **Torsion Angles:** For torsion angles, both FlexSBDD and our approach employ flow matching on the manifold of hypertorus, originally proposed for full-atom peptide design by Li et al. (2024). However, given the amino acid sequence in SBDD, we can explicitly address cases where certain residues have side-chain torsion angles with -rotation symmetry (e.g., of ASP). This is a more natural choice than FlexSBDD's method, which overlooks symmetry-induced angle period differences. For more details, see Section 3.2 and App. B.
  - **SDE Variants:** Both DYNAMICFLOW-ODE (ours) and FlexSBDD use ODEs to model transitions between apo and holo states and the ligand generation process. However, we also introduce an SDE variant to enhance robustness, with experimental results demonstrating that the DYNAMICFLOW-SDE variant outperforms the DYNAMICFLOW-ODE. For more details, refer to Section 3.3.
  - **Interaction Loss:** FlexSBDD models predict the vector field directly, while our approach predicts "clean" samples and reparameterizes them into vector fields. This allows us to introduce an interaction loss focused on atom distances, enhancing the learning of protein-ligand interactions from ground-truth data. Our experiments show that this interaction loss improves the model's understanding of these interactions and enhances the binding affinity of generated ligands.
- **Evaluation:** FlexSBDD assesses generated small molecule ligands based on QED, SA, Binding Affinity (measured by Vina), and profiles of protein-ligand interaction. We evaluate baselines and our methods from these perspectives, and also add an evaluation of how similar the generated pocket structures are to actual holo states by comparing pocket volume and RMSD. For details, see Section 4, especially Fig. 5, and Fig. 6.

The above differences underline our unique approach to incorporating protein dynamics in SBDD.

In addition to our focus on structure-based drug design, an alternative and widely utilized approach in drug design is **ligand-based drug design**. This method does not explicitly use the 3D structures of target proteins. Instead, it optimizes ligand molecules for specific properties, such as binding affinity to certain targets, using various algorithms. The ligands themselves are modeled, while target information is incorporated implicitly. We list several representative examples in the following. Xie et al. (2021) employed Markov chain Monte Carlo sampling (MCMC) on molecules with a GNN-based adaptive proposal model to optimize molecules towards multiple desired properties. Bengio et al. (2021) proposed a flow network for modeling distribution that is proportional to the rewards, from which the molecules can be sequentially sampled. Fu et al. (2022) proposed a differentiable scaffolding tree for molecular optimization. FREED (Yang et al., 2021) and FREED++ (Telepov et al.,

2023) utilized fragment-based molecule generation models combined with reinforcement learning algorithms, leveraging desired properties as rewards for designing molecules.

There are also extensive works focused on other tasks instead of (small molecule) drug design but also modeling protein-ligand complexes. Notably, the input conditions (i.e., known information) and the output goals (i.e., the components to be generated) differ from those of our task. Our focus is on structure-based drug design (SBDD) considering protein dynamics, where we start with the apo state (initial pocket structure) and aim to generate the holo state and binding ligands. In our case, detailed ligand information, including both topology graphs and 3D structures, is not provided. We will discuss the differences of these related works, using representative examples to illustrate them, as follows: **Pocket Design**: Zhang et al. (2023b) concentrates on pocket design where the topology graph and initial 3D structure of the ligand are provided and the goal is to generate a compatible pocket for binding. **Protein-Ligand Complex Structure Generation:** Gao et al. (2024) focuses on protein-ligand complex structure generation where the protein sequence and the topology graph (i.e., 2D graph) of the ligand molecule are provided and only their 3D structures need to be generated. **Pocket Representation Learning:** Nakata et al. (2023) focuses on pocket representation learning via pretraining on pseudo-ligand-pocket complexes instead of SBDD.

