# OpenReview forum: "Integrating Protein Dynamics into Structure-Based Drug Design via Full-Atom Stochastic Flows"
_ICLR.cc/2025/Conference — ICLR 2025 Poster_

### Official Review · Reviewer_vrce · 2024-10-21

**Soundness:** 3
**Presentation:** 3
**Contribution:** 3
**Rating:** 5
**Confidence:** 4

**Summary:**

The paper curate a dataset of apo and multiple holo states of protein-ligand complexes, simulated by molecular dynamics, and propose a full-atom flow model (and a stochastic version), named DynamicFlow, that learns to transform apo pockets and noisy ligands into holo pockets and corresponding 3D ligand molecules. The experimental results seem to demonstrate that the model significantly improves the inputs for SBDD methods and enables the generation of ligand molecules with high binding affinity.

**Strengths:**

1. The results indicate that the proposed method is effective in enhancing the binding ability between ligands and proteins, with the Vina score reflecting a strong binding affinity.​
2. The concept of incorporating dynamic adaptability into drug design is uncommon in the existing literature and demonstrates a high level of originality.

**Weaknesses:**

1. The results pertaining to the main proposed evaluation algorithm appear to align with previous studies. Several algorithms referenced are well established in the literature. The authors should:

- Clearly specify which algorithms are original and unique to this study.

- Explicitly indicate which algorithms are derived from existing works, rather than from the authors' own proofs.

2. The paper employs various professional terms and abbreviations; however, the backgrounds and specific definitions for these terms are not adequately clarified. For example, the term "stochastic full-atom flow" within the model lacks a clear explanation of its exact meaning and implementation methods.

3. Regarding the dynamic adaptation of ligands and proteins, although relevant mathematical models are provided, there is a lack of essential intuitive explanations. Additionally, the logical relationships and technical details of certain steps are not distinctly articulated.

4. The analysis and treatment of different states (apo and holo) in the dataset are not sufficiently explored. Moreover, the specific reasoning behind the ratio of the various states (apo to holo) remains unclear.

5. There is a notable absence of baseline methods presented. It would be beneficial to consider comparing a broader range of graph-based generative models to conduct a thorough evaluation of the model's performance.

6. Reviews of computer science papers typically encourage the inclusion of anonymous code, accompanied by straightforward and easily testable data. Furthermore, it is recommended to incorporate a Jupyter Notebook, facilitating readers' understanding of the method presented.

**Questions:**

What challenges does this method have in dealing with protein flexibility? How can we learn from the direction of the solution?

---

> ### Author Response · Authors · 2024-11-22
> **Response to Reviewer vrce (1/N)**
>
> Thank you for your feedback. Please see below for our responses to the comments.
>
> **Q1: "The results pertaining to the main proposed evaluation algorithm appear to align with previous studies. Several algorithms referenced are well established in the literature. The authors should: clearly specify which algorithms are original and unique to this study; explicitly indicate which algorithms are derived from existing works, rather than from the authors' own proofs."**
>
> A1:
> Our work introduces a novel setting in structure-based drug design (SBDD) where protein dynamics are considered, necessitating the creation of new evaluation algorithms.
>
> The evaluation algorithms derived from existing works include:
>
> - **Ligand property evaluation**: This includes metrics like QED, SA, Vina Score, Lipinski, logP, High Affinity, and Complete Rate. These are widely used in previous SBDD works, such as TargetDiff [1]. Please see Lines 476-484 for details and related references.
> - **RMSD**: This metric is commonly used to measure the structural difference between two structures and is frequently employed in prior research, such as AlphaFold2 [2]. In our work, it measures how closely our generated pocket structures resemble the real holo states.
> - **Protein-ligand non-covalent interaction profiling**: This is also utilized in previous works, e.g., [3,4].
>
> The evaluation algorithms original and unique to this study are:
>
> - **Cover ratio**: Based on RMSD, this metric demonstrates the diversity and detailed capability of our method to generate holo-like states. Details can be found on Lines 508-512.
> - **Pocket volume difference**: We propose using pocket volume differences to evaluate how generated pocket structures resemble real holo states. While the evaluation perspective is novel, the calculation of pocket volume employs well-established tools like POVME 3 [5].
> - **Binding Affinity evaluation considering protein dynamics**: Considering protein dynamics in SBDD, we extend this consideration to our evaluations. We assess all methods using flexible docking and predicted affinity based on DynamicBind [6], introducing a novel evaluation protocol. For more details, refer to Q7 & A7 for Reviewer FBxC.
>
> **References:**
>
> [1] Guan, J., Qian, W. W., Peng, X., Su, Y., Peng, J., & Ma, J. (2023). 3d equivariant diffusion for target-aware molecule generation and affinity prediction. ICLR 2023.
>
> [2] Jumper, J., Evans, R., Pritzel, A., Green, T., Figurnov, M., Ronneberger, O., ... & Hassabis, D. (2021). Highly accurate protein structure prediction with AlphaFold. Nature, 596(7873), 583-589.
>
> [3] Lee, J., Zhung, W., & Kim, W. Y. (2024). NCIDiff: Non-covalent Interaction-generative Diffusion Model for Improving Reliability of 3D Molecule Generation Inside Protein Pocket. arXiv preprint arXiv:2405.16861.
>
> [4] Zhang, Z., Shen, W. X., Liu, Q., & Zitnik, M. (2024). Efficient generation of protein pockets with PocketGen. Nature Machine Intelligence, 1-14.
>
> [5] Wagner, J. R., Sørensen, J., Hensley, N., Wong, C., Zhu, C., Perison, T., & Amaro, R. E. (2017). POVME 3.0: software for mapping binding pocket flexibility. Journal of chemical theory and computation, 13(9), 4584-4592.
>
> [6]  Lu, W., Zhang, J., Huang, W., Zhang, Z., Jia, X., Wang, Z., ... & Zheng, S. (2024). DynamicBind: Predicting ligand-specific protein-ligand complex structure with a deep equivariant generative model. Nature Communications, 15(1), 1071.
>
> **Q2: "The paper employs various professional terms and abbreviations; however, the backgrounds and specific definitions for these terms are not adequately clarified (e.g., the term, stochastic full-atom flow)."**
>
> A2:
> Thank you for highlighting this concern. The term "stochastic full-atom flow" combines three aspects: "stochastic," "full-atom," and "flow." Here’s what each term signifies:
>
> - "Flow" indicates that we use a flow model as our generative model.
> - "Full-atom" implies that our model explicitly captures atom-level protein-ligand interactions, as opposed to only modeling the protein pockets at the residue level.
> - "Stochastic" refers to the stochastic differential equation (SDE) variant of DynamicFlow, where stochasticity is introduced to enhance robustness.
>
> These meanings are intended to be intuitive and straightforward. However, as suggested, we will ensure they are more clearly defined in future versions of our paper. We have introduced biological terminologies in Section 1 (Introduction) and covered mathematical terms in Section 3.1 (Background and Preliminaries), where we provide sufficient explanations. If there are any other professional terms or abbreviations that need clarification, please let us know, and we will gladly offer more detailed explanations.

---

> ### Author Response · Authors · 2024-11-22
> **Response to Reviewer vrce (2/N)**
>
> **Q3: "Although relevant mathematical models are provided, there is a lack of essential intuitive explanations."**
>
> A3:
> We have indeed provided intuitive explanations for each mathematical component of our work. Flow models naturally model the transition between two distributions. Therefore, our framework intuitively employs flow models to represent the transition of pocket structures from the apo to the holo state, alongside the ligand generation process. The detailed mathematics in Section 3 specifically outlines how we model changes in pocket conformation and ligand during this process. If there are any questions regarding the intuitive explanations of specific mathematical parts, we are more than willing to discuss them further.
>
> **Q4: "The analysis and treatment of different states (apo and holo) in the dataset are not sufficiently explored. Moreover, the specific reasoning behind the ratio of the various states (apo to holo) remains unclear."**
>
> A4: We specify the dataset curation process in Appendix A, including the data source (see Line 813-830), holo and apo definition and the relevant treatments (see Line 833-842, Line 880-904). We keep the top 10 clusters (or fewer if the total number of clusters are smaller than 10) in every clustered complex MD data as different holo conformations (see Line 840-841). That is to say, each complex has one apo pocket structure predicted by alphafold2 and no more than 10 holo structures extracted from MD simulation. We also present some important figures for dataset visualizations and analysis. Figure 9 shows the number of comformation in dataset, from which we make clustering and define the holo structures. Figure 11 shows the distribution of molecular properties for ligands in holo conformations. To compare the apo and holo pockets in terms of binding affinity with ligands, we present the distribution of vina score and vina min for apo and holo complexes in Figure 12.
>
> **Q5: "It would be beneficial to consider comparing a broader range of graph-based generative models to conduct a thorough evaluation of the model's performance."**
>
> A5: In our study, we have already included several representative graph-based generative models as baselines. These models span different categories, such as autoregressive models and diffusion models. It's important to note that these baselines are typically limited to rigid pockets. Our work, however, introduces a novel and practical approach by considering protein dynamics in structure-based drug design. This is where our significant contribution lies, as we provide a suitable algorithm and model tailored to this new setting.
>
> **Q6: "Reviews of computer science papers typically encourage the inclusion of anonymous code, accompanied by straightforward and easily testable data."**
>
> A6: Upon acceptance of this paper, we will open-source both the code and the curated dataset, and offer a user-friendly interface for ease of use. We are also open to discussing further implementation details if needed. Please refer to Q6 & A6 of Reviewer gwDc for specific details on hyperparameters and model architectures.
>
> **Q7: "What challenges does this method have in dealing with protein flexibility? How can we learn from the direction of the solution?"**
>
> A7: We recognize that data scarcity is a significant barrier to accurately modeling flexible proteins. Although the MISATO dataset marks crucial progress, obtaining stable, long-term MD simulation data for complexes remains challenging due to high computational costs. Furthermore, advocating for the public release of more high-quality complex data is essential to overcoming this challenge.
>
> A current trend is to incorporate more prior knowledge rather than relying solely on data-driven approaches. For instance, we've introduced full-atom representation and interaction loss in protein dynamics modeling, which prevents the model from learning atom-level protein-ligand interactions too implicitly, potentially hindering its learning.
>
> In the future, a promising direction will be how to better incoporate physical rules into protein modeling. For example, how protein force-fields can be integrated in the modeling process to help models understand protein dynamics. A physical-informed architecture may offer a more optimal solution to tackle this challenge.

---

### Official Review · Reviewer_JxBu · 2024-10-26

**Soundness:** 3
**Presentation:** 3
**Contribution:** 3
**Rating:** 8
**Confidence:** 3

**Summary:**

Authors developed the 3D pocket conditioned generative model, named DynamicFlow, for small molecule drug discovery. Compared to existing 3D generative models that require HOLO pocket structures, the proposed model can generate protein-ligand binding HOLO structures from the APO protein structure predicted by protein structure prediction models such as AlphaFold.

**Strengths:**

1. The authors extend the 3D structure-based ligand binder design problem to an Apo pocket structure setting.
2. They demonstrate that the predicted protein Holo structures can be utilized as input structures for existing 3D molecular generation models.
3. The authors perform not only docking but also non-covalent interaction analysis. This will likely serve as a desirable experimental analysis design for future 3D molecular generation model works.

**Weaknesses:**

**Overall comment.**

The authors effectively address protein flexibility relative to small molecules using state-of-the-art flow matching techniques.
I'm not very interested in the 3D molecular generative models due to its unproven practicality; however, setting this bias aside, I consider this paper an important milestone in this field. However, questions remain regarding the quality of generated samples and the real-world applicability of the proposed methodology.

*\* The issues are sorted in the order they appeared in the manuscript.*

**Issue 1: Bias in the training data.** Page 8. Section 4. Data curation.

This work relies on training datasets constructed from MD simulations, which may lead the model to learn the simulated physics of MD rather than capturing real-world structural distributions. While various 3D generative models ([1-2]) utilized simulation datasets (e.g., CrossDocked2020), I think it would be a minor issue. For future work, authors might consider leveraging true experimental datasets PDBbind [3] to enhance reliability [4].

**Issue 2: Bias in the predicted pocket structure selection for analysis.** Page 9. Table 2, Page 10 Figure 6.

In the paper, authors selected the pocket structure ("our pocket") among the predicted pocket conformations based on Vina score, so the distribution of selected structures is biased from the training dataset. I suggest adding the analysis about randomly selected conformations, too. If the performance of SBDD methods (Table 2) or Volume distribution (Figure 6) are similar when using true holo structure and randomly selected structure, this would be strong evidence that the model has learned the distribution of holo structures.

**Issue 3: Questions about applicability in real-world applications.** Page 16. Line 835. "we locate residues within a cutoff distance of 7Å around each ligand and extract them from the 100-frame MD results."

As I understand it, this study define the pocket using the atom coordinate informations of known active binders in holo structures.
However, I wonder whether these pocket definitions are directly applicable to Apo protein structures generated in AlphaFold.
For high usability, the pockets should be easily defined, e.g., defining pocket using the center and radius (or box length) in an Apo structure.
This process has been used in existing 3D generative models [1-2], but this is because they do not account for the flexibility of the pockets.
If I misunderstood the process, let me know.


---
**Reference.**
1. Peng, Xingang, et al. "Pocket2mol: Efficient molecular sampling based on 3d protein pockets." International Conference on Machine Learning. PMLR, 2022.
2. Guan, Jiaqi, et al. "3d equivariant diffusion for target-aware molecule generation and affinity prediction." arXiv preprint arXiv:2303.03543 (2023).
3. Wang, Renxiao, et al. "The PDBbind database: methodologies and updates." Journal of medicinal chemistry 48.12 (2005): 4111-4119.
4. Zhung, Wonho, Hyeongwoo Kim, and Woo Youn Kim. "3D molecular generative framework for interaction-guided drug design." Nature Communications 15.1 (2024): 2688.

**Questions:**

1. **Page 8, Table 1.** The proposed models and baselines are distribution learning-based models. Therefore, QED, SA, Lipinski, logP should be similar to Reference ligands. (No $\uparrow$ or $\downarrow$.)
2. What is the generation time scale?

---

> ### Author Response · Authors · 2024-11-22
> **Response to Reviewer JxBu**
>
> Thank you for your positive feedback. Please see below for our responses to the comments.
>
> **Q1: "This work relies on training datasets constructed from MD simulations, which may lead the model to learn the simulated physics of MD rather than capturing real-world structural distributions. While various 3D generative models utilized simulation datasets (e.g., CrossDocked2020), I think it would be a minor issue. For future work, authors might consider leveraging true experimental datasets PDBbind to enhance reliability."**
>
> A1: Thanks for your suggestion. Indeed, simulated data may induce bias. Nevertheless, MD simulation is usually more reliable than docking methods as used in CrossDocked2020, though it is more computationally expensive. And the MISATAO dataset is actually sourced from PDBBind. Besides, we have filtered the MD simulated data to further enhance reliability. Please refer to Appendix A for details of data processing. We would like to follow your suggestion to simultaneously leverage more experimental data to improve our work as a future work.
>
> **Q2: "Bias in the predicted pocket structure selection for analysis."**
>
> A2: Thanks for your suggestion. Our work aims at exploring more holo-like states given an initial conformation of pocket. Thus, we select pockets yielding the best results based on Vina score, with the intention of identifying more suitable holo-like states for ligand design.
>
> Additionally, we calculated the volume for randomly selected pockets and compared the volume differences between apo states and our generated pockets (via DynamicFlow-ODE and DynamicFlow-SDE) against real holo states for each target. We have reported their mean and median in the table below. More details concerning the volume calculation and a specific example can be found in Appendix H of the revised version (marked in blue).
>
> |  | Volume difference from holo states |  |
> |---|---|---|
> |  | Avg. ± Std. | Med. |
> | Apo States | 83.84 ± 61.20 | 71.20 |
> | Our Pocket (DynamicFlow-ODE) | **50.08  ± 35.05** | **41.75** |
> | Our Pocket (DynamicFlow-SDE) | 68.56 ±  55.51 | 59.20 |
>
> The results confirm that our methods successfully discover holo-like pockets. Furthermore, we assessed binding affinity via flexible docking on randomly selected and all generated ligands to thoroughly evaluate performance. Our methods outperformed all baselines under these conditions. For more details, please refer to Q7 & A7 for Reviewer FBxC.
>
> **Q3: "Questions about applicability in real-world applications."**
>
> A3:
> In real-world applications with only apo structures available, the pocket region can be effectively defined using a geometric center and radius. This method is efficient because the pocket areas in apo and holo states are generally similar, even if their conformations differ. Due to the lack of a clearly defined pocket boundary in receptor proteins, our model can leverage this ambiguity. Both ligand-centric and receptor-centric methods are expected to produce similarly defined pocket regions for SBDD tasks, allowing our model to adapt and apply these definitions to apo protein structures generated by tools like AlphaFold, unconstrained by the rigid boundaries present in holo structures.
>
> In the few cases where protein conformational changes might be significantly large, experts can select or define the pocket in the apo state. This expert input can propose diverse pocket definitions, facilitating the design of potential ligands using our model.
>
> **Q4: "The proposed models and baselines are distribution learning-based models. Therefore, QED, SA, Lipinski, logP should be similar to Reference ligands. (No ↑ or ↓.)"**
>
> A4: Thanks for pointing this out. We agree that from a distribution learning standpoint, generated ligands with more similar statistics to reference ligands indicate a superior model. In our work, we followed conventions from other studies, like TargetDiff, by using "↑" or "↓" to denote preferences in drug design. We will include a note on this in the revised version. From the distribution learning perspective, our methods outperform others across nearly all metrics, as they most closely approximate the reference molecules in terms of property statistics.
>
> **Q5: "What is the generation time scale?"**
>
> A5: We benchmark the inference time of baselines and our methods for generating 10 ligand molecules given the same pocket on 1 Tesla V100-SXM2-32GB. The default number of function evaluations (NFE) is 1000 for TargetDiff and IPDiff and 100 for our method.
>
> |  | Time (s) | Default NFE |
> |---|---|---|
> | Pocket2Mol | 980 | N/A |
> | TargetDiff | 156 | 1000 |
> | TargetDiff* | 154 | 1000 |
> | IPDiff | 334 | 1000 |
> | IPDiff* | 343 | 1000 |
> | DynamicFlow-ODE | 35 | 100 |
> | DynamicFlow-SDE | 36 | 100 |
>
> As the results show, our methods are capable of generating high-quality ligands while simultaneously modeling protein dynamics at a fast speed, demonstrating a significant advantage in computational efficiency.

---

> ### Comment · Reviewer_JxBu · 2024-11-23
> **Official Comment of  by Reviewer JxBu**
>
> Thank you for response. Most of my concern is addressed. I'll maintain my score (8).

---

> > ### Author Response · Authors · 2024-11-24
> >
> > Thank you once again for your positive feedback and kind support!
> >
> > We sincerely appreciate your recognition of our efforts and contributions.

---

### Official Review · Reviewer_7Up8 · 2024-11-01

**Soundness:** 3
**Presentation:** 3
**Contribution:** 2
**Rating:** 6
**Confidence:** 3

**Summary:**

The paper proposes Dynamic Flow, a flow-matching-based method designed for structure-based drug discovery (SBDD) with a focus on protein flexibility. Specifically, Dynamic Flow models the mappings between apo (unbound) and holo (bound) protein conformations, as well as between an ideal normal distribution and the actual ligand conformation distribution. To effectively train the model on meaningful mappings between apo and holo conformations, the authors introduce a new dataset that includes molecular dynamics (MD)-modeled apo-holo protein conformations paired with ligand conformers.

**Strengths:**

1) The curated dataset, where each apo protein pocket is mapped to multiple holo pockets, is novel and well-suited for the SBDD task involving protein flexibility.
2) The paper is clearly written and supported by well-designed figures, which enhance comprehension of the proposed method.

**Weaknesses:**

1, The curated dataset in the paper is slightly different from the commonly used benchmark dataset in SBDD which are bindingMOAD and crossdock2020 and also the datasize is smaller compared with those two, I’m curious why not start with crossdock2020 and BindingMOAD before moving into new dataset?

2, About baseline: As far as I know, there is another work on SBDD with protein flexibility via flow matching that published or submitted ahead of this work FlexSBDD[1] on Neurips 2024, I would suggest author to benchmark with their number on vina results and illustrate the novelty and improvement compare with this previous work.

3, The paper lists “atom-level SE(3)-equivariant geometrical message-passing layers and residue-level Transformer layers” as contributions; however, there is no ablation study showing the impact of these architectural changes on model performance. Providing an ablation to assess their effects on sample quality (e.g., Vina score) or computational efficiency (e.g., FLOPs) would strengthen the evidence of these improvements.

Ref:
[1]FlexSBDD: Structure-Based Drug Design with Flexible Protein Modeling

**Questions:**

I was wondering, how did the author select the 50 test pockets other than having no overlap with the training set? Are they uniformly sampled from the whole dataset

---

> ### Author Response · Authors · 2024-11-22
> **Response to Reviewer 7Up8 (1/N)**
>
> Thank you for your feedback. Please see below for our responses to the comments.
>
> **Q1: "Why not start with CrossDock2020 and BindingMOAD before moving into new dataset?"**
>
> A1: CrossDocked2020 enhances PDBBind with docking and filtering, but MD simulations offer greater accuracy and provide multiple valid holo states rather than just one. BindingMOAD resembles PDBBind as it comprises protein-ligand crystal structures. The MISATO dataset, however, includes molecular dynamics simulations for approximately 20,000 experimental protein-ligand complexes and is slightly larger than BindingMOAD. Since part of our work is focused on discovering holo-like states in the context of structure-based drug design (SBDD), a dataset enriched with diverse holo states is preferred. Thus, starting with a dataset like MISATO aligns well with our objectives.
>
>
> **Q2: Comparison with FlexSBDD published on NeurIPS 2024.**
>
> A2:
> Thank you for bringing FlexSBDD to our attention. According to ICLR Review Guidelines (https://iclr.cc/Conferences/2025/ReviewerGuide), contemporaneous papers—those published within four months of our submission—need not be compared. FlexSBDD became publicly available on September 29, 2024, just days before our deadline on October 1, 2024, showing we are independent works. We acknowledge FlexSBDD and our work both integrate protein flexibility or dynamics into SBDD but present differences in various aspects. Although their code has not been open-sourced, we will cite and discuss their work in future revisions.
>
> Key distinctions between our work and FlexSBDD include:
>
> - **Motivation**: FlexSBDD primarily seeks to incorporate protein flexibility into SBDD for optimizing complex structures and ligands. However, it overlooks the role of thermodynamic fluctuations that govern protein flexibility and conformational shifts, leading to diverse conformations with different ligands. In contrast, our work delves into the physics underlying these dynamics. We illustrate this by examining the DFG-in and DFG-out states of Abl kinase, emphasizing our motivation to integrate comprehensive protein dynamics into SBDD, beyond merely addressing flexibility.
>
> - **Data**: FlexSBDD derives most of its apo data by augmenting holo data through relaxation/sidechain repacking. In contrast, we use AlphaFold2 to predict our apo data, potentially resulting in greater conformational changes. Additionally, our holo states are diverse, providing multiple states for each protein-ligand pair through molecular dynamics simulations, enabling a more thorough exploration of pocket conformational changes. This aligns with our motivation.
>
> (continued on the next session)

---

> ### Author Response · Authors · 2024-11-22
> **Response to Reviewer 7Up8 (2/N)**
>
> - **Methodology**:
>   - **Protein Modeling**: FlexSBDD employs a residue-level model for the protein pocket, whereas we utilize both residue-level and atom-level models simultaneously, leveraging atom37 mapping. This approach allows us to more precisely capture protein-ligand interactions at the atomic level, enhancing the accuracy and detail of our modeling.
>   - **Ligand Modeling**: On the ligand side, we construct flow models for atom positions, atom types, and bond types simultaneously in an end-to-end manner. In contrast, FlexSBDD does not incorporate bond modeling within its flow model, instead generating bonds through empirical post-processing rules. Our approach allows for more integrated and cohesive modeling of ligand structures.
>   - **Discrete Variable Modeling**: FlexSBDD uses continuous vectors to represent discrete variables (i.e., atom types) and utilize standard flow matching for continuous variables, employing "norm" for self-normalization to mimic probabilities. This introduces a lack of rigor due to the inference gap created by "norm." Conversely, we apply rigorous discrete flow matching using continuous-time Markov chains (CTMC) to model both atom and bond types, ensuring a more precise and theoretically robust representation. For detailed mathematical insights, see Section 3.1 and Lines 291-309.
>   - **Torsion Angles**: For torsion angles, both FlexSBDD and our approach employ flow matching on the manifold of hypertorus, originally proposed for full-atom peptide design by Li et al. [1]. However, given the amino acid sequence in SBDD, we can explicitly address cases where certain residues have side-chain torsion angles with -rotation symmetry (e.g.,  of ASP). This is a more natural choice than FlexSBDD's method, which overlooks symmetry-induced angle period differences. For more details, see Lines 270-288 and Appendix B.
>   - **SDE Variants**: Both DynamicFlow-ODE (ours) and FlexSBDD use ODEs to model transitions between apo and holo states and the ligand generation process. However, we also introduce an SDE variant to enhance robustness, with experimental results demonstrating that the DynamicFlow-SDE variant outperforms the DynamicFlow-ODE. For more details, refer to Section 3.3.
>   - **Interaction Loss**: FlexSBDD models predict the vector field directly, while our approach predicts "clean" samples and reparameterizes them into vector fields. This allows us to introduce an interaction loss focused on atom distances, enhancing the learning of protein-ligand interactions from ground-truth data. Our experiments show that this interaction loss improves the model's understanding of these interactions and enhances the binding affinity of generated ligands.
> - **Evaluation**: FlexSBDD assesses generated small molecule ligands based on QED, SA, Binding Affinity (measured by Vina), and profiles of protein-ligand interaction. We evaluate baselines and our methods from these perspectives, and also add an evaluation of how similar the generated pocket structures are to actual holo states by comparing pocket volume and RMSD. For details, see Lines 508-514, Figure 5, and Figure 6.
>
> These differences underline our unique approach to incorporating protein dynamics in SBDD.
>
> **References:**
>
> [1] Li, Jiahan, et al. "Full-Atom Peptide Design based on Multi-modal Flow Matching." ICML 2024.

---

> ### Author Response · Authors · 2024-11-22
> **Response to Reviewer 7Up8 (3/N)**
>
> **Q3: "There is no ablation study showing the impact of these architectural changes (i.e., atom-level SE(3)-equivariant geometrical message-passing layers and residue-level Transformer layers) on model performance. Providing an ablation to assess their effects on sample quality (e.g., Vina score) or computational efficiency (e.g., FLOPs) would strengthen the evidence of these improvements."**
>
> A3:
> Based on your suggestion, we conducted ablation studies to evaluate the impact of different architectural components on model performance. We implemented a baseline denoted as "w/o residue-level Transformer", which uses only atom-level SE(3)-equivariant geometrical message-passing layers. In this setup, atom-level output features are aggregated into residue-level features without employing a residue-level Transformer for further extraction, and these aggregated features are used to predict the residue frames’ translation, rotation, and torsion angles.
>
> Additionally, we developed a baseline referred to as "w/o atom-level EGNN", which transforms the atom-level protein-ligand complex graph into a heterogeneous graph, where each node represents either a residue (with C-alpha coordinates, rotation vectors, and torsion angles as input features) or a ligand atom. In this variant, since we do not explicitly reconstruct the full atom representation of the pocket, the atom interaction loss is not applied.
>
> The results are shown in the following table:
>
> |  | Vina Score | QED | SA |
> |---|---|---|---|
> | DynamicFlow-ODE | -7.28 ± 1.98 | 0.53 ± 0.20 | 0.61 ± 0.14 |
> | w/o interaction loss | -6.76 ± 1.39 | 0.54 ± 0.22 | 0.60 ± 0.15 |
> | w/o residue-level Transformer | -6.23 ± 1.68 | 0.53 ± 0.22 | 0.59 ± 0.14 |
> | w/o atom-level EGNN | -6.02 ± 1.63 | 0.54 ± 0.19 | 0.64 ± 0.13 |
> | DynamicFlow-SDE | -7.65 ± 1.59 | 0.53 ± 0.15  | 0.53 ± 0.17 |
> | w/o interaction loss | -7.00 ± 1.15 | 0.48 ± 0.21 | 0.56 ± 0.16 |
> | w/o residue-level Transformer | -6.50 ±  1.22 | 0.52 ± 0.16 | 0.56  ± 0.14 |
> | w/o atom-level EGNN | -6.13 ±  1.31 | 0.49 ± 0.19 | 0.60  ± 0.16 |
>
> The results indicate that our proposed architecture significantly enhances binding affinity and is vital for effectively modeling protein-ligand interactions and protein dynamics.
>
> Both variants ("w/o residue-level Transformer" and "w/o atom-level EGNN") are more computationally efficient due to their reduced model sizes. However, despite using both residue-level and atom-level models, our method maintains acceptable inference speed because our flow model can generate high-quality ligand molecules in fewer steps. (Refer to Q5 & A5 for Reviewer JxBu for a comparison of inference time between the baselines and our methods.)
>
>
> **Q4: "How did the author select the 50 test pockets other than having no overlap with the training set? "**
>
> A4: We ensured no overlap by verifying that for each holo pocket in the test set, the PM-score against any holo pocket in the training set is less than 0.95. The PM-score quantifies binding-site similarity using structural descriptors like residue nature and interatomic distances, calculated via PocketMatch [1]. We plan to explore additional similarity measures and data splitting methods in future work.
>
> **References:**
>
> [1] Nagarajan, D., & Chandra, N. (2013, February). PocketMatch (version 2.0): A parallel algorithm for the detection of structural similarities between protein ligand binding-sites. In 2013 National Conference on Parallel Computing Technologies (PARCOMPTECH) (pp. 1-6). IEEE.

---

> > ### Comment · Reviewer_7Up8 · 2024-11-24
> >
> > Thank you for your response. My concerns are addressed. I'll raise my score to 6.

---

### Official Review · Reviewer_FBxC · 2024-11-02

**Soundness:** 2
**Presentation:** 2
**Contribution:** 2
**Rating:** 6
**Confidence:** 3

**Summary:**

The authors propose a new dataset of apo/holo proteins and a new model to solve the mapping task between apo protein pockets and holo protein pockets/ligand complexes.

**Strengths:**

- A new dataset derived from the MISATO dataset and AlphaFold Protein Structure Database, which includes 5,692 complexes with 46,235 holo-ligand conformations and corresponding apo structures.
- A new task to map apo protein pockets to holo protein pockets/ligand complexes.
- A new generative model, DynamicFlow, with a stochastic variation, which is based on the combination of discrete and continuous Flow Matchings, that simultaneously generates a ligand and adjusts a protein pocket.

**Weaknesses:**

- The dataset collection process is not very clear; see the questions below.
- The dataset is simulational, which makes it less representative than the PDB.
- The paper lacks the baselines for joint pocket and ligand generation or the motivation for their absence  [1, 2, 3, 4].
- The evaluation pipeline seems unfair and may be misleading: baseline models are designed to work with holo-state pockets, whereas, in the experiment, they are utilized to generate ligands for apo-state pockets.
- Unlike the baselines, DynamicFlow-generated ligands are evaluated inside the generated pockets, which also affects the comparison. From our point of view, a valid way to compare different settings for baselines and DYNAMICFLOW is to use MD trajectories-based methods, such as MMPBSA [5].
- Moreover, the fact that a ligand binds well to a generated pocket may not imply that the real affinity is good. The validity of the generated pockets is not assessed. For example, the AF PLDDT may be used.
- Table 2. shows that the diffusion methods work better for the ligand prediction given the generated pocket than the proposed model itself, which increases the concern about the lack of comparison of the model’s architecture with the other known models

**Questions:**

## Questions and remarks
1. How did you transition from 19437 proteins to 16972 complexes? How are ligands selected, and why was the original number of proteins reduced?
2. What does "100-frame" mean? Every 100th frame of the MD trajectory? Or 100 frames from each MD simulation? If the second is true, how were the 100 frames selected from the MD trajectory?
3. Do you filter the protein-ligand complexes depending on the average RMSD throughout the trajectory? Please explain Figure 10 b.
4. Please add the DynamicsFlow models to Table 2 for easier comparison.
5. FREED [6] and FREED++ [7] are not cited.
6. Line 214: string repetition.
7. line 848: mistake (are can).

## Closing remarks

Overall, I find the paper important for drug design. The idea that the pocket changes upon the introduction of the ligand is well physically motivated. The generative model itself is impressive, as it combines various flow-matching modules. However, the experimental evaluation raises serious concerns about the validity of the comparison with the baselines. I would consider raising my score if the authors propose a better evaluation strategy or resolve concerns with the current evaluation.

[1] Zhang, Z., Lu, Z., Zhongkai, H., Zitnik, M., & Liu, Q. (2023). Full-atom protein pocket design via iterative refinement. Advances in Neural Information Processing Systems, 36, 16816-16836.

[2] Gao, B., Jia, Y., Mo, Y., Ni, Y., Ma, W. Y., Ma, Z. M., & Lan, Y. Self-supervised Pocket Pretraining via Protein Fragment-Surroundings Alignment. In The Twelfth International Conference on Learning Representations.

[3] Nakata, S., Mori, Y., & Tanaka, S. (2023). End-to-end protein–ligand complex structure generation with diffusion-based generative models. BMC bioinformatics, 24(1), 233.

[4] Huang, L., Xu, T., Yu, Y., Zhao, P., Chen, X., Han, J., ... & Zhang, H. (2024). A dual diffusion model enables 3D molecule generation and lead optimization based on target pockets. Nature Communications, 15(1), 2657.

[5] Wang, E., Sun, H., Wang, J., Wang, Z., Liu, H., Zhang, J. Z., & Hou, T. (2019). End-point binding free energy calculation with MM/PBSA and MM/GBSA: strategies and applications in drug design. Chemical reviews, 119(16), 9478-9508

[6] Yang, S., Hwang, D., Lee, S., Ryu, S., & Hwang, S. J. (2021). Hit and lead discovery with explorative rl and fragment-based molecule generation. Advances in Neural Information Processing Systems, 34, 7924-7936.

[7] Telepov, A., Tsypin, A., Khrabrov, K., Yakukhnov, S., Strashnov, P., Zhilyaev, P., ... & Kadurin, A. FREED++: Improving RL Agents for Fragment-Based Molecule Generation by Thorough Reproduction. Transactions on Machine Learning Research.

---

> ### Author Response · Authors · 2024-11-22
> **Response to Reviewer FBxC (1/N)**
>
> Thank you for your detailed feedback. Please see below for our responses to the comments.
>
> **Q1: "How did you transition from 19437 proteins to 16972 complexes? How are ligands selected, and why was the original number of proteins reduced?"**
>
> The MISATO dataset is curated using 19,443 protein-ligand complexes data from PDBbind (release 2022). According to the author, structures from PDBbind were excluded whenever non-standard ligand atoms or inconsistencies in the protein starting structures were encountered, resulting in 16972 complexes for MD simulation. We further exclude complexes with oligopeptide ligands, resulting in 12,695 complexes for further processing (see L833). Detailed data processing procedures are provided in Appendix A.
>
> **Q2: "What does "100-frame" mean? Every 100th frame of the MD trajectory? Or 100 frames from each MD simulation? If the second is true, how were the 100 frames selected from the MD trajectory?"**
>
> A2: The MISATO dataset collects 100 snapshots for each protein-ligand complex from the 8 ns MD trajectory with systematic sampling. We then cluster the 100 snapshots for each complex using an RMSD threshold of 1 Å (see Line 893).
>
> **Q3: Do you filter the protein-ligand complexes depending on the average RMSD throughout the trajectory? Please explain Figure 10 b.**
>
> A3: Yes, we filter the clustered holo structures based on the average RMSD_Ligand of the MD trajectory, applying a 3 Å threshold (see Line 966-967). A large RMSD_Ligand suggests potential unreliability in the MD trajectory of the protein-ligand complex, rendering the data questionable. The RMSD_Ligand measures the root-mean-square deviation of the ligand after aligning the protein with its native structure.
>
> Figure 10 b shows the change in the number of complexes along our data processing procedures. Stage A represents the original MISATO dataset with 16,972 complexes. At Stage B, We filter out complexes where ligands are peptides, resulting in 12,695 complexes (see Line 886-887). At Stage C, we align proteins in our data and AlphaFold Database by sequence and filter out the unsuccessful cases, resulting in 7,528 complexes (see Line 945-958 for details). At stage D, we remove the data with RMSD_Ligand smaller than 3 Å, resulting in 5,692 complexes (see Line 964-967 for details).
>
> **Q4: "The dataset is simulational, which makes it less representative than the PDB."**
>
> A4: The MISATO dataset originates from PDBbind, where the protein-ligand structures are experiment-based, as indicated in Line 877. While these structures provide a solid experimental foundation, it's important to recognize that protein-ligand complexes are inherently dynamic, and their holo states are not singular. Incorporating molecular dynamics (MD) simulations enhances the dataset by introducing additional dynamic conformational information, which aids the model in exploring a broader range of valid holo states. Furthermore, our processed dataset can be utilized for other significant tasks, such as conformational sampling of protein-ligand complexes, thereby making a valuable contribution to the research community.

---

> ### Author Response · Authors · 2024-11-22
> **Response to Reviewer FBxC (2/N)**
>
> **Q5: "The paper lacks the baselines for joint pocket and ligand generation or the motivation for their absence."**
>
> A5: Thanks for highlighting the related works [1,2,3,4]. We have cited and discussed these in Appendix L in the newly-updated revision. **Although these works pertain to protein-ligand complex modeling, they address distinct tasks.**
>
> Our focus is on structure-based drug design (SBDD) considering protein dynamics, where we start with the apo state (initial pocket structure) and aim to generate the holo state and binding ligands. In our case, detailed ligand information, including both topology graphs and 3D structures, is not provided.
>
> [1] concentrates on pocket design where the topology graph and initial 3D structure of the ligand are provided and the goal is to generate a compatible pocket for binding.
>
> [2] focuses on protein-ligand complex structure generation where the protein sequence and the topology graph (i.e., 2D graph) of the ligand molecule are provided and only their 3D structures need to be generated.
>
> [3] focuses on pocket representation learning via pretraining on pseudo-ligand-pocket complexes instead of SBDD.
>
> [4] represents a standard SBDD method with rigid-pocket input. Although molecular dynamics were mentioned in this work, they refer to dynamics induced by the forward process of the diffusion model. In our work, we have compared our methods with various similar baselines [5,6]. They are all diffusion-based SBDD methods with rigid-pocket input, with slight differences in models or algorithms.
>
> **References:**
>
> [1] Zhang, Z., Lu, Z., Zhongkai, H., Zitnik, M., & Liu, Q. (2023). Full-atom protein pocket design via iterative refinement. Advances in Neural Information Processing Systems, 36, 16816-16836.
>
> [2] Gao, B., Jia, Y., Mo, Y., Ni, Y., Ma, W. Y., Ma, Z. M., & Lan, Y. Self-supervised Pocket Pretraining via Protein Fragment-Surroundings Alignment. In The Twelfth International Conference on Learning Representations.
>
> [3] Nakata, S., Mori, Y., & Tanaka, S. (2023). End-to-end protein–ligand complex structure generation with diffusion-based generative models. BMC bioinformatics, 24(1), 233.
>
> [4] Huang, L., Xu, T., Yu, Y., Zhao, P., Chen, X., Han, J., ... & Zhang, H. (2024). A dual diffusion model enables 3D molecule generation and lead optimization based on target pockets. Nature Communications, 15(1), 2657.
>
> [5] Guan, J., Qian, W. W., Peng, X., Su, Y., Peng, J., & Ma, J. (2023). 3d equivariant diffusion for target-aware molecule generation and affinity prediction. ICLR 2023.
>
> [6] Huang, Z., Yang, L., Zhou, X., Zhang, Z., Zhang, W., Zheng, X., ... & Yang, W. (2024). Protein-ligand interaction prior for binding-aware 3d molecule diffusion models. ICLR 2024.
>
> **Q6: "The evaluation pipeline seems unfair and may be misleading: baseline models are designed to work with holo-state pockets, whereas, in the experiment, they are utilized to generate ligands for apo-state pockets."**
>
> A6: Our work represents a pioneering effort to integrate protein dynamics into structure-based drug design (SBDD), introducing a novel experimental setting. Given its unique and innovative nature, achieving a completely fair comparison with existing baseline models is inherently challenging. Our experimental setup is intentionally crafted to simulate scenarios where complete holo-state structures may not be readily available. This highlights the necessity of developing solutions that can effectively generate ligands using apo-state pockets, thus addressing an often overlooked yet critical aspect of the drug design process.

---

> ### Author Response · Authors · 2024-11-22
> **Response to Reviewer FBxC (3/N)**
>
> **Q7: A valid way to compute different settings for baselines and DynamicFlow is to use MD trajectories-based methods, such as MMPBSA.**
>
> A7: We agree that incorporating molecular dynamics in evaluation is important. However, relying on MD trajectory-based methods such as MMGBSA or MMPBSA can be highly cumbersome. These approaches employ different solvation models and require extensive computational resources, with MD simulations often taking months to complete for thousands of systems. Therefore, as an alternative, we opted for a deep-learning-based approach for flexible docking and scoring to achieve reliable and scalable evaluation.
>
> Specifically, for each generated ligand designed by the baselines and our methods, we employ DynamicBind [1], a geometric deep generative model tailored for "dynamic docking", to generate 10 protein-ligand complex structures. DynamicBind also includes a model that predicts an "affinity" score, which estimates the negative logarithm of the binding affinity in concentration units. We then calculate the weighted average of these predicted binding affinities to derive the final "affinity" score, where a higher "affinity" score indicates better binding potential.
>
> For each target, we assess the affinity of a randomly selected generated ligand (Single), the highest affinity among 10 generated ligands (Best over 10), and the best affinity across all 100 generated ligands (Best over all). We report the mean, standard deviation, and median of these affinities across 50 targets. The results are summarized as follows:
>
> |  | Single |  | Best over 10 |  | Best over all |  |
> |---|---|---|---|---|---|---|
> |  | Avg. ± Std. | Med. | Avg. ± Std. | Med. | Avg. ± Std. | Med. |
> | Pocket2Mol | 3.64 ± 1.26 | 3.31 | 4.90 ± 1.15 | 4.81    | 5.70 ± 1.22 | 5.68 |
> | TargetDiff | 6.00 ± 1.14 | 6.19 | 7.30 ± 0.70 | 7.46 | 7.81 ± 0.71 | 7.91 |
> | TargetDiff* | 6.19 ± 0.97 | 6.38 | 7.16 ± 0.94 | 7.54 | 7.64 ± 0.73 | 7.79 |
> | IPDiff | 6.15 ± 1.14 | 6.45 | 7.05 ± 0.79 | 7.18 | 7.68 ± 0.90 | 7.82 |
> | IPDiff* | 5.96 ± 1.31 | 5.83 | 7.10 ± 1.09 | 7.14 | 7.63 ± 0.97 | 7.72 |
> | DynamicFlow-ODE | **6.46 ± 1.00** | **6.69** | 7.40 ± 0.94 | 7.62 | 7.91 ± 0.90 | 8.07 |
> | DynamicFlow-SDE | 6.21 ± 1.19 | 6.09 | **7.53 ± 0.86** | **7.67** | **7.95 ± 0.83** | **8.12**  |
>
> The results demonstrate that our methods outperform all baseline models across all evaluation settings. This highlights the strength of our approach in designing ligands with high binding affinity.
>
> **References:**
>
> [1] Lu, W., Zhang, J., Huang, W., Zhang, Z., Jia, X., Wang, Z., ... & Zheng, S. (2024). DynamicBind: Predicting ligand-specific protein-ligand complex structure with a deep equivariant generative model. Nature Communications, 15(1), 1071.
>
>
> **Q8: "Please add the DynamicsFlow models to Table 2 for easier comparison." "Table 2 shows that the diffusion methods work better for the ligand prediction given the generated pocket than the proposed model itself."**
>
> A8: Table 2 presents the performance of baseline models using either apo states or the pocket structures generated by DynamicFlow as inputs. Specifically, the entry labeled "Pocket2Mol" represents Pocket2Mol with apo states as input, while "Pocket2Mol + Our Pocket" represents Pocket2Mol utilizing pocket structures generated by DynamicFlow. The results highlight that DynamicFlow effectively discovers more appropriate holo states, thereby facilitating the design of ligands with high binding affinity.
>
> The results show that some baselines outperform our proposed model in ligand prediction when using the generated pockets, which aligns with our expectations. This is largely because the baselines have the advantage of more extensive training data, whereas our method is specifically designed to discover pocket structures that closely resemble actual holo states. Consequently, with an accurate holo-like state, baselines might surpass our model in performance. However, this does not undermine the utility of our model; rather, it underscores its capability to discover pocket structures that closely align with real-world scenarios.

---

> ### Author Response · Authors · 2024-11-22
> **Response to Reviewer FBxC (4/N)**
>
> **Q9: "The validity of the generated pockets is not assessed. For example, the AF PLDDT may be used."**
>
> A9: Assessing the validity of the generated pocket structures poses a significant challenge. We have evaluated the pocket volume distribution, and our findings indicate that the generated pocket structures exhibit volumes more akin to MD-simulated holo pockets compared to apo pockets. The AF pLDDT metric is unsuitable for our assessment for two primary reasons: (i) AF pLDDT provides a measure of per-residue local confidence specifically for AF-predicted structures and cannot be computed for structures not predicted by AF; and (ii) it does not account for the presence of ligands. We acknowledge the importance of validating generated pocket structures and propose developing a related benchmark as future work.
>
> **Q10: FREED and FREED++ are not cited.**
>
> A10: FREED [1] and FREED++ [2] are ligand-based drug design methods, differing from structure-based drug design approaches. They utilize fragment-based molecule generation models combined with reinforcement learning algorithms, leveraging desired properties as rewards for designing molecules. Importantly, the 3D structures of pockets are not inputs to these models. In our revision, we will cite these methods and discuss the distinctions between their approach and ours.
>
> **References:**
>
> [1] Yang, S., Hwang, D., Lee, S., Ryu, S., & Hwang, S. J. (2021). Hit and lead discovery with explorative rl and fragment-based molecule generation. Advances in Neural Information Processing Systems, 34, 7924-7936.
>
> [2] Telepov, A., Tsypin, A., Khrabrov, K., Yakukhnov, S., Strashnov, P., Zhilyaev, P., ... & Kadurin, A. FREED++: Improving RL Agents for Fragment-Based Molecule Generation by Thorough Reproduction. Transactions on Machine Learning Research.
>
> **Q11: Typos: "Line 214: string repetition" and "line 848: mistake (are can)".**
>
> A11: Thanks for pointing these out. We have fixed the typos in the revision.

---

### Official Review · Reviewer_gwDc · 2024-11-03

**Soundness:** 2
**Presentation:** 2
**Contribution:** 3
**Rating:** 6
**Confidence:** 2

**Summary:**

Summary:
1. This paper tackles the problem of flexible proteins in small molecule structure based drug discovery to account for protein's conformational changes during binding.
2. The authors propose a full-atom model based on continuous flow matching for pocket residues' translation, rotation, torsional angles and ligand molecule's atom position) and discrete flow matching for atom and bond types of ligand molecules.
3. They finally present a stochastic version of their flow matching objective for increased robustness.

**Strengths:**

Strengths:
1. The protein flexibity problem is critical in drug discovery with no good computationally efficient insilico methods. Hence, the proposed method (if works as advertised) could be paradigm shifting.
2. The flow matching components in the paper are generally well explained.
3. The structural features of protein and ligands are carefully considered and appropriately modeled.

**Weaknesses:**

Weaknesses:
1. I have concerns about the reproducibility of this work with no code or the curated dataset (mentioned as a key contribution) not provided. Moreover access to the code would have helped understanding the complex workflow in the paper better.
2. In table 2, where are the results for dynamic flow?
3. In figure 1, what are protein and ligand embeddings, where are they computed in the proposed workflow and how are they being used in complex graph and ligand graph respectively? Given the number of components in the workflow, I would suggest including an aggerated workflow figure with end to end pipeline starting from the apo state input to the molecule and holo state output, complete with the final flow matching loss.
4. The overall loss for this work is unclear, while individual losses for structural features for protein and ligand are provided, how are they aggregated is not mentioned.
5. Hyperparameter and other architectural details are also not provided.

**Questions:**

I found certain parts of the paper difficult to follow. Particularly in section 3.4, the details of  each of the $\phi_i$ are mising. Are they all EGNNs? I also struggled to understand how this parametrization relates to the final flow matching objective to be used. Furthermore, some parts in this section feel hand-wavy. For example, how were the hidden states used to predict atom positions and atom/bond types. Similarly in residue-level transformer, L412, final updated frames were used as predictions - for what? It was again unclear how torsion angles were predicted based on final residue level hidden states.

---

> ### Author Response · Authors · 2024-11-22
> **Response to Reviewer gwDc (1/N)**
>
> Thank you for your feedback. Please see below for our responses to the comments.
>
> **Q1: About reproducibility.**
>
> A1: To help researchers better understand our framework, we will open-source both the code and curated dataset upon acceptance of this paper and provide a user-friendly interface. Additionally, we will provide further details about the model architecture and hyperparameters, as discussed in Q6 & A6, to enhance understanding.
>
> **Q2: "In Table 2, where are the results for DynamicFlow?"**
>
> A2: . Table 2 shows the performance of the rigid-pocket SBDD methods with our refined pocket conformation (i.e., holo pocket structures generated by DynamicFlow). More specifically, the entry "TargetDiff" corresponds to TargetDiff with apo pockets as input, and the entry "TargetDiff + Our pocket" corresponds to TargetDiff with holo states generated by DynamicFlow as input. This experiment shows that the holo states discovered by DynamicFlow might serve as better inputs for the rigid-pocket SBDD methods and improve their performance when real holo pockets are not available. We will include more descriptions about this in the revision to enhance clarity.
>
> **Q3: "In Figure 1, what are protein and ligand embeddings, where are they computed in the proposed workflow and how are they being used in complex graph and ligand graph respectively?"**
>
> A3: Figure 1 shows different holo conformations of Abl kinase with corresponding binding ligands as an example to illustrate the motivation of our work. In Figure 3, protein and ligand embeddings are derived from the encodings of protein atom features and ligand atom and bond features, respectively, through an embedding layer (i.e., learnable linear transformation). The protein atom feature contains its atom37 representation and residue type. (Atom37 is an all-atom representation of proteins where each heavy atom corresponds to a given position in a 37-dimensional array. This mapping is non amino acid specific, but each slot corresponds to an atom of a given name. Note that atom37 is widely used in protein modeling [1]) We concatenate the one-hot encodings of these two features (whose dimensions are 37 and 20, respectively) to derive the protein atom encodings (whose dimension is 57). We use the one-hot encoding of the atom type as the ligand atom encoding. We only consider explicitly modeling "C, N, O, F, P, S, Cl, Br" in ligands, so the dimension is 8. For ligand bond types, we consider "non-bond, single, double, triple, aromatic", so the dimension of ligand bond encoding is 5. The protein and ligand atom features are used as the initial node features in the complex graph. And the ligand atom and bond features are used as the initial node and edge features, respectively, in the ligand graph. The above encodings are common in modeling proteins and small molecules. We will include more details about this in the revision to improve clarity.
>
> **References:**
>
> [1] Jumper, J., Evans, R., Pritzel, A., Green, T., Figurnov, M., Ronneberger, O., Tunyasuvunakool, K., Bates, R., Žídek, A., Potapenko, A. and Bridgland, A., 2021. Highly accurate protein structure prediction with AlphaFold. Nature, 596(7873), pp.583-589.
>
> **Q4: "Given the number of components in the workflow, I would suggest including an aggregated workflow figure with end to end pipeline starting from the apo state input to the molecule and holo state output, complete with the final flow matching loss."**
>
> A4: Thanks for your suggestion. We will include a more comprehensive illustration of the overall workflow to promote understanding in the revision.
>
> **Q5: "The overall loss for this work is unclear, while individual losses for structural features for protein and ligand are provided, how are they aggregated is not mentioned."**
>
> A5: There are 7 individual losses: 4 continuous flow matching losses for residue frames' translation (Equation 5), rotation (Equation 7), torsion angles (Equation 8), and ligand atom position (same as Equation 5), 2 discrete flow matching losses for ligand atom and bond types (Equation 14), and interaction loss (Equation 18). They are first averaged across all residues or atoms in a training sample and then simply weighted summed with weights: 2.0, 1.0, 1.0, 4.0, 1.0, 1.0, 0.5.

---

> ### Author Response · Authors · 2024-11-22
> **Response to Reviewer gwDc (2/N)**
>
> **Q6: Hyperparameter and other architectural details.**
>
> A6: The hyperparameters about the training loss are provided in Q5 & A5 and directly in Equation 18. We use AdamW [1] as the optimizer with learning rate 0.0002, beta1 0.95, and beta2 0.999.
> Gamma $\gamma$ controls the stochasticity of the stochastic flow (see Equations 19, 20, and 21). We use 2.0, 0.005, 1.0, 2.0 as the values of gamma $\gamma$ for residue frames' translation, rotation, torsion angles, and ligand atom positions.
>
> Our model consists of an atom-level SE(3)-equivariant graph neural network and a residue-level Transformers. The number of total parameters is 15.9 M. The total estimated model parameter size is 63.401 MB.
>
> We include more details about the model architecture as follows:
>
> |  | Layer name (which also indicate its function) | Number of layers |
> |---|---|---|
> | Atom-level Model | Protein atom embedding layer | 1 |
> |  | Ligand atom embedding layer  | 1 |
> |  | Ligand bond embedding layer  | 1 |
> |  | Time embedding layer | 1 |
> |  | EGNN block | 6 |
> |  | Ligand atom type prediction head  | 1 |
> |  | Ligand bond type prediction head  | 1 |
> | Residue-level Model | Protein residue embedding layer  | 1 |
> |  | Time embedding layer | 1 |
> |  | Transformer block with IPA (invariant point attention)  | 4 |
> |  | Torsion angle prediction head | 1 |
>
>
>
>
> **Q7: Details of each $\phi_i$ in Section 3.4 and how this parameterization relates to the final flow matching objective.**
>
> A7: Each $\phi$ in Section 3.4 is an SE(3)-equivariant graph neural network. As introduced in Section 3.4 (especially Line 407-413) and Figure 3, these SE(3)-equivariant graph neural networks are atom-level and their outputs are used to predict ligand atom types and positions (which are further used to compute the flow matching losses for ligand atom/bond types and ligand atom positions) and serve as inputs of the residue-level Transformer. The outputs of residue-level Transformer are used to compute flow matching losses for protein residue frames' translation, rotation, and torsion angles. We hope our explanation helps you understand how these EGNNs relate to the final flow matching objectives.
>
> **Q8: "How were the hidden states used to predict atom positions and atom/bond types? Similarly in residue-level transformer, L412, final updated frames were used as predictions - for what? It was again unclear how torsion angles were predicted based on final residue level hidden states."**
>
> A8: The final SE(3)-equivariant features of ligand atom are directly used as predicted ligand atom positions without any further transformation. The final SE(3)-invariant features of ligand atom/bond are used to predict atom/bond types by a linear layer. The related details of the residue-level Transformer can also be found in FrameDiff [2]. Specifically, the SE(3)-invariant hidden states, translation and rotation of residue frames are updated after each Transformer block. The translation and rotation of the final frame are viewed as the final prediction without any further transformation or post process. The hidden states are used to predict the torsion angles by a linear layer, where each torsion angle of each residue is a scalar value. These operations are very simple and common.
>
> **References:**
>
> [1] Loshchilov, I., 2017. Decoupled weight decay regularization. arXiv preprint arXiv:1711.05101.
>
> [2] Yim, J., Trippe, B.L., De Bortoli, V., Mathieu, E., Doucet, A., Barzilay, R. and Jaakkola, T., 2023. SE (3) diffusion model with application to protein backbone generation. ICML 2023.

---

> > ### Comment · Reviewer_gwDc · 2024-11-26
> >
> > Dear Authors,
> >
> > Thank you for updating the manuscript to add further clarifications. I still look forward to your code however, in light of additional details provided, I am willing to raise my score to 6.

---

> ### Author Response · Authors · 2024-11-24
>
> Thank you for taking the time and effort to evaluate our submission!
>
> We have just updated the revision again to include a newly added comprehensive illustration of the overall workflow, as you suggested. Please refer to Appendix D and Figure 14 for further details.

---

### Public Comment · ~Kiwoong_Yoo1 · 2025-06-13
**When is the code released?**

Hi, this is exciting work, when will the code be released? thanks!!

---

### Meta-Review · Area_Chair_U49S · 2024-12-22

**Metareview:**

The paper introduces a new dataset of apo/holo protein structures and proposes a flow matching model to map apo pocket structures to possible holo full-atom structures in conjunction with a generated ligand’s bound structure.

The reviewers appreciated the construction of the dataset, the relevance of the task of simultaneous pocket conformation and ligand generation for structure based drug discovery, as well as the accessible presentation and writing of the paper.

They also had concerns regarding the curation process of the dataset including possible bias, the choice of baselines for joint pocket-ligand pose generation, and the evaluation process.

The authors provided a thorough rebuttal where they clarified several points including the curation of the dataset, the choice of the baselines, the potential bias of the dataset and evaluations and they provided new experiments including new evaluations of the affinity using pretrained DynamicBind.

The rebuttal convinced most of the reviewers. The remaining outstanding issue, raised by vrce who leaned towards rejection, is the novelty of the proposed method.

The AC agrees with vrce that the technical novelty is limited but given the novelty of application and significance of the achieved results, the AC defers to the majority of reviewers which favorably rated the paper and therefore recommends acceptance.

**Additional Comments On Reviewer Discussion:**

The paper was reviewed by a panel of five expert reviewers with diverse expertise from the application to the proposed technique. Four out of five reviewers eventually rated the paper for acceptance while one reviewer remained unconvinced despite the rebuttal with the main criticism being the lack of novelty. The AC agrees with the low technical novelty but believes that alone is not ground for rejection as the other concerns have been successfully rectified.

---

### Decision · Program_Chairs · 2025-01-22

Accept (Poster)